



# Measurements of carbonyl compounds around the Arabian Peninsula indicate large missing sources of acetaldehyde

Nijing Wang[1], Achim Edtbauer[1], Christof Stönner[1], Andrea Pozzer[1], Efstratios Bourtsoukidis[1], Lisa Ernle[1],

Dirk Dienhart[1], Bettina Hottmann[1], Horst Fischer[1], Jan Schuladen[1], John N. Crowley[1], Jean-Daniel Paris[2],

Jos Lelieveld[1, 3], Jonathan Williams[1, 3]

[1] Air Chemistry Department, Max Planck Institute for Chemistry, Hahn-Meitner-Weg 1, 55128 Mainz, Germany

[2] Laboratoire des Sciences du Climat et de l'Environnement, LSCE/IPSL, CEA-CNRS-UVSQ, Université Paris-Saclay, Gif-sur-Yvette, France

[3] Energy, Environment and Water Research Center, The Cyprus Institute, Nicosia, Cyprus

*Correspondence to:* Nijing Wang (nijing.wang@mpic.de)

**Abstract**

Volatile organic compounds (VOCs) were measured around the Arabian Peninsula using a research vessel during the AQABA campaign (Air Quality and Climate Change in the Arabian Basin) from June to August 2017. In this study we examine carbonyl compounds (CxHyO), measured by a proton transfer reaction mass spectrometer (PTR-ToF-MS), and present both a regional concentration distribution and a budget assessment for these key atmospheric species. Among the aliphatic carbonyls, acetone had the highest mixing ratios in most of the regions traversed, varying from 0.43 ppb over the Arabian Sea to 4.5 ppb over the Arabian Gulf, followed by formaldehyde (measured by Hantzsch monitor, 0.82 ppb over the Arabian Sea and 3,8 ppb over the Arabian Gulf) and acetaldehyde (0.16 ppb over the Arabian Sea and 1.7 ppb over the Arabian Gulf). Unsaturated carbonyls (C4-C9) varied from 10 to 700 ppt during the campaign, and followed similar regional mixing ratio dependence as aliphatic carbonyls, which were identified as oxidation products of cycloalkanes over polluted areas. An empirical method based on hydrocarbon ratios was applied to investigate the photochemical source strength of the aliphatic carbonyls. While the distribution and relative concentration enhancements of the C3-C8 aliphatic carbonyls could be explained by this method, that of acetaldehyde could not. A smaller but still significant discrepancy was found when comparing measurements to global chemistry-transport model (EMAC) results, with the model underestimating the measured acetaldehyde mixing ratio up to an order of magnitude. Implementing a photolytically driven marine source of acetaldehyde significantly improved the agreement between measurements and model, particularly over the remote regions (e.g. Arabian Sea). However, the newly introduced acetaldehyde source was still insufficient to describe the observations over the most polluted regions (Arabian Gulf and Suez), where model underestimation of primary emissions and biomass burning events are possible reasons.



## 1 Introduction

Carbonyl compounds (aldehydes and ketones) can be released into the air directly from a variety of primary biogenic and anthropogenic sources. These include biomass burning (Holzinger et al., 1999;Holzinger et al., 2005;Koss et al., 2018), fossil fuel combustion (Reda et al., 2014;Huang et al., 2018) including vehicles (Erickson et al., 2014;Dong et al., 2014), industrial solvent use (Kim et al., 2008), and natural sources including plants and plankton (Zhou and Mopper, 1997;Warneke et al., 1999; Jacob et al., 2002;Fall, 2003;Williams et. al., 2004; Bourtsoukidis et al., 2014). However, secondary production via the atmospheric oxidation of hydrocarbons is considered to be more important for many carbonyl compounds including acetone and acetaldehyde (Jacob et al., 2002; Millet et al., 2010).

Carbonyls have several important roles in the atmosphere. They form as stable intermediates directly after hydrocarbon oxidation by OH, $O_3$ or $NO_3$, when the peroxy radicals initially formed react with each other (permutation reactions) or with NO. Their production is linked to tropospheric ozone formation (Carlier et al., 1986) and their loss, through oxidation and photolysis, is an important source of free radicals (HOx) in the dry upper troposphere (Colomb et al., 2006). Carbonyls serve as precursors of peroxyacetyl nitrates (PANs) which are important atmospheric NOx reservoir species (Finlayson-Pitts and Pitts, 1997;Edwards et al., 2014;Williams et al., 2000). Carbonyl compounds are also important for the growth of atmospheric particles (Kroll et al. 2005) thereby indirectly impacting the Earth's radiative balance. The atmospheric lifetimes of carbonyl compounds varies considerably, from less than one day for acetaldehyde (Millet et al., 2010) to more than 15 days for acetone (Jacob et al., 2002;Khan et al., 2015) in terms of tropospheric mean lifetime. A multiday lifetime means that carbonyl compounds can impact the air chemistry on local, regional and even hemispheric scales. The numerous primary and secondary sources of carbonyl compounds, as well as their multiple loss routes (photolysis, OH, $NO_3$ and $O_3$ oxidation) makes budget assessments difficult.

The most predominant atmospheric carbonyl compounds besides formaldehyde are acetaldehyde and acetone. They have been reported to vary from a few hundred ppt in remote areas (Warneke and de Gouw, 2001;Lewis et al., 2005;White et al., 2008;Colomb et al., 2009;Read et al., 2012;Sjostedt et al., 2012;Tanimoto et al., 2014;Hornbrook et al., 2016) to several ppb in urban and polluted areas (Dolgorouky et al., 2012;Guo et al., 2013;Stoeckenius and McNally, 2014;Sahu et al., 2017;Sheng et al., 2018). Generally, secondary photochemical formation from various precursors is the main source in those regions. However, several recent studies have shown that acetaldehyde mixing ratios in both the remote marine boundary layer and the free troposphere could not be explained by known photochemistry as implemented in various atmospheric chemistry models, which consistently underestimated the measurements by an order of magnitude or more (Singh et al., 2003;Read et al., 2012;Wang et al., 2019). Several potential additional acetaldehyde sources have been proposed including new hydrocarbon oxidation mechanisms, aerosol related sources and oceanic sources. One possible source of acetaldehyde in the remote marine boundary layer is oceanic emission from the photo degradation of colored dissolved organic matter (CDOM) in sea-surface water, where acetaldehyde could be produced together with other low molecular weight carbonyl compounds (Kieber et al., 1990;Zhou and Mopper, 1997;Sinha et al., 2007;Dixon et al., 2013). Nevertheless, due to both limited airborne and seawater measurements of acetaldehyde, the importance of oceanic emission is still under debate (Millet et al., 2010;Wang et al., 2019). In order to better understand the atmospheric budgets of acetaldehyde (and the other carbonyl compounds), it is informative to analyze a dataset of multiple carbonyl compounds in both polluted and clean environments, with influence from marine emissions, varying particulate loadings, and high rates of oxidation as shown in Figure 1, which demonstrates the main formation pathways of acetaldehyde during this campaign.

During the shipborne research campaign AQABA, carbonyl compounds were continuously measured by PTR-ToF-MS onboard a research vessel that circumnavigated the Arabian Peninsula. During the campaign, chemically distinct air masses were sampled,



which had been influenced by primary emissions of hydrocarbons and inorganic pollutants (NO$_x$, SO$_2$ and CO) from petroleum
industries and marine transport (Bourtsoukidis et al., 2019;Celik et al., 2019), by pollution from urban areas (Pfannerstill et al.,
2019), and clean marine influenced air. It is a unique dataset of carbonyl compounds encompassing starkly different environmental
conditions from a region with few (or none) available in-situ measurements to date.
In this study, we provide an overview of carbonyl compound mixing ratios (aliphatic, unsaturated and aromatic) over the
Mediterranean Sea, Suez, Red Sea, Arabian Sea and Arabian Gulf. Using an empirical method based on measured hydrocarbon
precursors, we have analyzed the relative importance of the photochemical sources for the carbonyl compounds observed. The
analysis is then extended to include sources and transport by using a global model EMAC (5th generation European Centre –
Hamburg general cir-culation model, **E**CHAM5 coupled to the modular earth submodel system, **M**ESSy, applied to **a**tmospheric
**c**hemistry). Model measurement differences are investigated in both clean and polluted regions, with particular emphasis on
acetaldehyde.
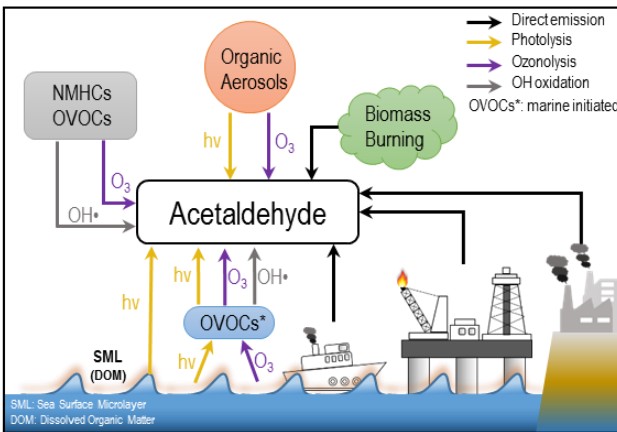
Figure 1. Diagram of possible sources and formation pathways of acetaldehyde during the AQABA campaign.

## 88 2 Methods

### 89 2.2 AQABA campaign

The AQABA campaign was conducted onboard the research vessel Kommandor Iona (KI) from the end of June to the end of
August 2017. The ship started from Southern France, proceeded across the Mediterranean, through the Suez Canal, around the
Arabian Peninsula into the Arabian Gulf and on to Kuwait, thereafter returning along the same route. Five laboratory containers
were loaded onto the vessel, containing multiple gas and particle phase measurement instruments as well as a weather station.

### 94 2.3 PTR-ToF-MS

#### 95 2.3.1 Sampling and instrument set-up

A high-flow inlet (stainless steel tubing, 0.2 m diameter, 5.5 m tall and 3 m above the top of the containers and the front deck) was
installed at the front of the ship where the laboratory containers were located. A high flow of air (approximately 10 m$^3$min$^{-1}$) was
drawn through the inlet which provided a common attachment point for sub-sampling lines for all gas-phase measurement
instruments. An air flow of 5 standard L min$^{-1}$ for the first leg and 3.5 standard L min$^{-1}$ for the second leg was pumped into the



VOC container through an ½" (O.D. = 1.27cm) FEP (fluorinated ethylene propylene) tubing (about 10 m long) insulated and heated
to 50-60 °C. A PTFE (polytetrafluoroethylene) filter was placed at the beginning of the inlet to prevent insects, dust and particles
entering the instruments. Every 2-5 days, the filter was replaced depending on the degree of pollution encountered.  Inside the
VOC instrument container, the PTR-ToF-MS (8000, Ionicon Analytik GmbH Innsbruck, Austria) sampled a sub-flow at 80-100
sccm through 1/8" (0.3175 cm) FEP tubing (~ 10 m in length, insulated and heated to 60 °C) from the main fast air flow and then
to the instrument's PEEK (polyether ether ketone) inlet which was likewise heated to 60 °C. The inlet system was shared with total
OH reactivity measurement (Pfannerstill et al., 2019).
The working principle of PTR-MS has been described in detail in previous studies (Lindinger et al., 1998;Ellis and Mayhew,
2013;Yuan et al., 2017). In brief, $H_3O^+$ primary ions are generated in the ion source, and then drawn into the drift tube where they
interact with sampled ambient air. Inside the drift tube, VOCs with a proton affinity greater than that of $H_2O$ (691 kJ $mol^{-1}$) are
protonated by proton transfer from $H_3O^+$. The resulting secondary ions are transferred to the detector, in this case a time-of-flight
mass spectrometer with mass resolution around 3500 for the first leg and 4500 for the second leg at mass 96amu. An internal
standard of trichlorobenzene ($C_6H_3Cl_3$) was continuously introduced into the instrument to ensure accurate mass calibration. Every
minute a spectrum with mass range (m/z) 0-450 was generated. The data reported in this study are all at 1 minute resolution unless
otherwise specified.

### 2.3.2    Instrument characterization

The instrument background was determined every three hours for 10 minutes with synthetic air. 4-point calibrations were
performed five times during the whole campaign using a standard gas mixture (Apel-Riemer Environmental inc., Broomfield, USA)
containing 14 compounds (methanol, acetonitrile, acetaldehyde, acetone, dimethyl sulfide, isoprene, methyl vinyl ketone,
methacrolein, methyl ethyl ketone, benzene, toluene, xylene, 1,3,5-trimethylbenzene and α-pinene). It has been previously reported
that the sensitivity of some compounds measured by PTR-MS are humidity dependent (de Gouw and Warneke, 2007). As the
relative humidity (RH) was expected to be high and varying (marine boundary layer with occasional desert air influence), humidity
calibration was combined with 4-point calibration by humidifying the gas mixture at different levels from 0% - 100% RH.

### 2.3.3    Data analysis

The data were initially processed by the PTR Analyzer software (Müller et al., 2013) to identify and integrate the peaks. After
obtaining the raw data (counts per second for each mass identified), a custom-developed python-based program was used to further
process the data to final mixing ratios. For compounds present in the standard gas cylinder, interpolated sensitivities based on the
five in-campaign calibrations were applied to derive the mixing ratios; while mixing ratios of the other masses were calculated by
using a proton transfer reaction rate constant ($k_{PTR}$) of $2.0 \times 10^{-9}$ $cm^3$ $s^{-1}$. The uncertainty associated with the mixing ratios of the
calibrated compounds was around 6-17% (see Table S1). For the mixing ratios derived by assuming $k_{PTR}$, the accuracy was around
±50% (Zhao and Zhang, 2004). The detection limit (LOD) was calculated from the background measurement with 3 times the
standard deviation (3σ), 52 ± 26 ppt for acetaldehyde, 22 ± 9 ppt for acetone and 9 ± 6 ppt for methyl ethyl ketone (MEK) (Table
S1).
In this study, we have interpreted ion masses with the exact masses corresponding to $C_nH_{2n}O$, $C_nH_{2n-2}O$ and $C_nH_{2n-8}O$ as aliphatic,
unsaturated and aromatic carbonyls, respectively (see exact protonated m/z in Table S2). Carbonyl compounds with a carbon
number three and above can be either aldehydes or ketones, which are not distinguishable with PTR-ToF-MS using $H_3O^+$ as the
primary ion. However, laboratory experiments have shown that protonated aldehydic ions with carbon atoms more than three tend





to lose a H$_2$O molecule and fragment to other masses (Buhr et al., 2002;Spanel et al., 2002). Moreover, although both ketones and
aldehydes can be produced via atmospheric oxidation processes, ketones tend to have longer atmospheric lifetimes than aldehydes
as mentioned in the introduction. Therefore, signals on the exact mass of carbonyl compounds from the PTR-ToF-MS are expected
to be dominated by ketones, particularly in regions remote from the sources.
**2.4 Meteorological data and other trace gases**
The meteorological data were obtained by using a commercial weather-station (Sterela) which monitored wind speed, wind
direction, relative humidity (RH), temperature, speed of the vessel, and GPS etc. The actinic flux was measured by a spectral
radiometer. Non methane hydrocarbons (NMHC) mixing ratios were measured by a gas chromatograph with flame ionization
detector (GC−FID), for a detailed instrumental description see Bourtsoukidis et al. (2019). Formaldehyde mixing ratios were
determined by a modified and optimized version of the commercially available AL4021 (Aero-Laser, Germany), which utilizes
the Hantzsch technique (Stickler et al., 2006). Methane and carbon monoxide (CO) levels were monitored by a cavity ring-down
spectroscopy analyzer (Picarro G2401). Ozone was measured with an absorption photometer (Model 202 Ozone Monitor, 2B
Technologies, Boulder, Colorado). Due to the potential interference from sampling our own ship exhaust in which carbonyl
compounds may be present (Reda et al., 2014), a filter was applied to the data set based on the wind direction and NO$_x$, SO$_2$ and
ethene levels.
**2.5 Model simulations**
The EMAC (ECHAM5/MESSy Atmospheric Chemistry) model was used to simulate atmospheric mixing ratios of several
carbonyl compounds along the cruise track covered during the AQABA campaign. The EMAC model is an atmospheric chemistry-
general circulation model simulating the process of tropospheric air by considering processes which could influence trace gases
mixing ratios, such as transport, chemistry, interaction with ocean/land, dry deposition and so on (Pozzer et al., 2007;Pozzer et al.,
2012;Lelieveld et al., 2016). The model applied in this study is a combination of the 5$^{th}$ generation of European Centre Hamburg
general circulation model (ECHAM5) (Roeckner et al., 2006) and the 2$^{nd}$ version of Modular Earth Submodel System (MESSy2)
(Jöckel et al., 2010), where a comprehensive chemistry mechanism MOM (Mainz Organic Mechanism) was deployed (Sander et
al., 2019). The model configuration in the study is the same as the model applied in Bourtsoukidis et al. (2020) in the resolution of
T106L31 (i.e. ~ 1.1° × 1.1° horizontal resolution and , 31 vertical hybrid pressure levels up to 10 hPa) and the time resolution of
10 minutes. The measurement data of PTR-ToF-MS were averaged to 10-minute resolution to match the model data resolution for
further comparison.
**3   Results and discussion**
**3.1 Overview of carbonyl compounds**
Around the Arabian Peninsula, the mixing ratios of individual carbonyl compounds varied over a wide range, from tens of ppt to
ppb levels. In this study, we divided the dataset geographically into eight regions (Figure 2, middle graph) to classify and
characterize the primary and secondary origins of carbonyl compounds. The regional delineations were: the Mediterranean Sea
(MS), Suez, Red Sea North (RSN), Red Sea South (RSS), Gulf of Aden (GA), Arabian Sea (AS), Gulf of Oman (GO) and Arabian
Gulf (AG), the same as those described by Bourtsoukidis et al. (2019). Figure 2 shows the abundance of aliphatic, aromatic and
unsaturated carbonyl compounds (carbonyls) for each region. Generally, aliphatic carbonyls were present at much higher mixing
ratios than aromatic and unsaturated carbonyls, with smaller carbonyl compounds (formaldehyde, acetaldehyde, C3 and C4
carbonyls) dominating the distribution. The mixing ratios of aliphatic carbonyls decreased dramatically from C5 carbonyls with





increasing carbon number. The box plots (Figure 2) also show that carbonyl compounds were measured at higher mixing ratios
and were more variable over Suez region and the Arabian Gulf. The abundance of carbonyl compounds varied markedly from
region to region with highest and lowest values found in the Arabian Gulf and the Arabian Sea, respectively. Table 1 shows the
mean, standard deviation and the median values for carbonyls in each region. In the following sections, each class of carbonyl
compounds are investigated in greater detail.

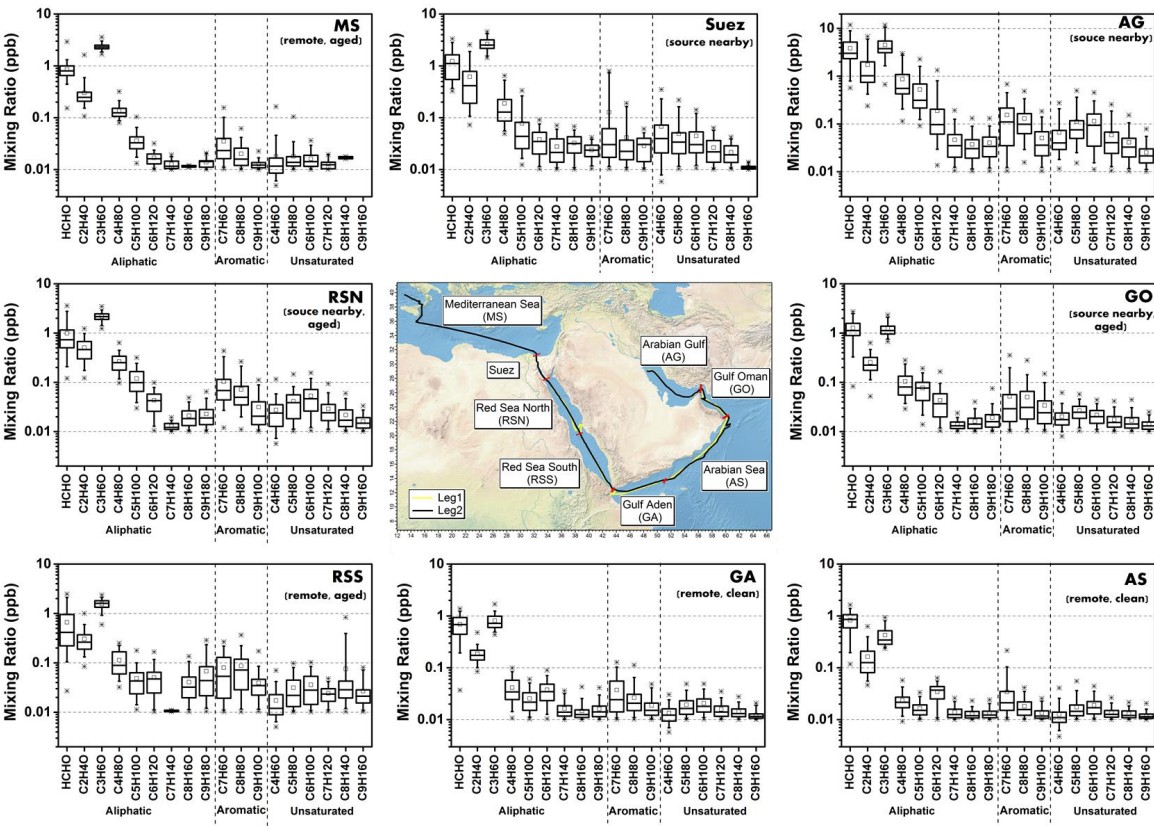


Figure 2. Overview of mixing ratios for aliphatic, aromatic and unsaturated carbonyl compounds (CxHyO). The boxes represent
25% to 75% of the data with the central line and square indicating the median and the mean values, respectively. The whiskers
show data from 5% to 95% and stars were drawn for the minimum and maximum data points within 1% to 99% of the dataset.
Within brackets under the region acronyms the main characteristics of the air masses are indicated, based on variability-lifetime
results (b factor) from Bourtsoukidis et al. (2019) and acetone mixing ratios in this study. The data used for map plotting was from
public domain GIS data found on the Natural Earth web site (http://www.naturalearthdata.com) and was read into Igor using the
IgorGIS XOP beta.








Table 1. Mean, standard deviation (SD) and median mixing ratios of aliphatic, unsaturated and aromatic carbonyls in different
regions.

| | | Aliphatic CCs | | | | | | | | |
| --- | --- | --- | --- | --- | --- | --- | --- | --- | --- | --- |
| | | HCHO | CH3CHO | C3H6O | C4H8O | C5H10O | C6H12O | C7H14O | C8H16O | C9H18O |
| MS | mean | 0.86 | 0.30 | 2.37 | 0.14 | 0.04 | 0.02 | 0.01 | 0.01 | 0.01 |
| | SD | 0.41 | 0.25 | 0.37 | 0.05 | 0.02 | 0.01 | 0.00 | 0.00 | 0.00 |
| | median | 0.80 | 0.25 | 2.32 | 0.12 | 0.03 | 0.02 | 0.01 | 0.01 | 0.01 |
| S | mean | 1.23 | 0.62 | 2.64 | 0.19 | 0.08 | 0.04 | 0.03 | 0.03 | 0.02 |
| | SD | 0.76 | 0.58 | 0.77 | 0.15 | 0.08 | 0.02 | 0.02 | 0.02 | 0.01 |
| | median | 1.11 | 0.42 | 2.52 | 0.13 | 0.04 | 0.04 | 0.02 | 0.03 | 0.02 |
| RSN | mean | 0.99 | 0.51 | 2.17 | 0.27 | 0.12 | 0.04 | 0.01 | 0.02 | 0.02 |
| | SD | 0.78 | 0.26 | 0.45 | 0.11 | 0.07 | 0.02 | 0.00 | 0.01 | 0.01 |
| | median | 0.73 | 0.46 | 2.17 | 0.25 | 0.10 | 0.04 | 0.01 | 0.02 | 0.02 |
| RSS | mean | 0.66 | 0.31 | 1.56 | 0.11 | 0.05 | 0.05 | 0.01 | 0.04 | 0.07 |
| | SD | 0.62 | 0.17 | 0.38 | 0.06 | 0.03 | 0.03 | 0.00 | 0.03 | 0.07 |
| | median | 0.40 | 0.26 | 1.60 | 0.09 | 0.04 | 0.05 | 0.01 | 0.03 | 0.04 |
| GA | mean | 0.69 | 0.19 | 0.81 | 0.04 | 0.03 | 0.04 | 0.02 | 0.02 | 0.02 |
| | SD | 0.33 | 0.08 | 0.27 | 0.02 | 0.01 | 0.02 | 0.01 | 0.01 | 0.01 |
| | median | 0.68 | 0.17 | 0.72 | 0.03 | 0.02 | 0.04 | 0.01 | 0.01 | 0.01 |
| AS | mean | 0.82 | 0.16 | 0.43 | 0.02 | 0.02 | 0.03 | 0.01 | 0.01 | 0.01 |
| | SD | 0.35 | 0.12 | 0.18 | 0.01 | 0.01 | 0.01 | 0.00 | 0.00 | 0.00 |
| | median | 0.86 | 0.13 | 0.34 | 0.02 | 0.02 | 0.04 | 0.01 | 0.01 | 0.01 |
| GO | mean | 1.27 | 0.26 | 1.33 | 0.10 | 0.08 | 0.04 | 0.01 | 0.02 | 0.02 |
| | SD | 0.59 | 0.12 | 0.40 | 0.06 | 0.04 | 0.03 | 0.00 | 0.01 | 0.01 |
| | median | 1.13 | 0.22 | 1.12 | 0.08 | 0.08 | 0.04 | 0.01 | 0.01 | 0.02 |
| AG | mean | 3.83 | 1.73 | 4.50 | 0.87 | 0.52 | 0.19 | 0.05 | 0.04 | 0.04 |
| | SD | 2.55 | 1.61 | 2.40 | 0.71 | 0.48 | 0.25 | 0.04 | 0.03 | 0.03 |
| | median | 3.02 | 1.02 | 3.77 | 0.56 | 0.31 | 0.10 | 0.04 | 0.03 | 0.03 |

















Table 1. Continued

|  |  | Aromatic CCs | | | Unsaturated CCs | | | | | |
|---|---|---|---|---|---|---|---|---|---|---|
|  |  | C7H6O | C8H8O | C9H10O | C4H6O | C5H8O | C6H10O | C7H12O | C8H14O | C9H16O |
| MS | mean | 0.04 | 0.02 | 0.01 | 0.02 | 0.02 | 0.02 | 0.01 | 0.02 | - |
|  | SD | 0.03 | 0.01 | 0.00 | 0.03 | 0.02 | 0.01 | 0.00 | 0.00 | - |
|  | median | 0.02 | 0.02 | 0.01 | 0.01 | 0.01 | 0.01 | 0.01 | 0.02 | - |
| S | mean | 0.13 | 0.04 | 0.03 | 0.07 | 0.05 | 0.05 | 0.03 | 0.02 | 0.01 |
|  | SD | 0.23 | 0.05 | 0.01 | 0.08 | 0.05 | 0.04 | 0.02 | 0.01 | 0.00 |
|  | median | 0.03 | 0.02 | 0.03 | 0.04 | 0.03 | 0.03 | 0.02 | 0.02 | 0.01 |
| RSN | mean | 0.10 | 0.07 | 0.03 | 0.03 | 0.04 | 0.05 | 0.03 | 0.02 | 0.02 |
|  | SD | 0.10 | 0.06 | 0.03 | 0.02 | 0.03 | 0.03 | 0.02 | 0.01 | 0.01 |
|  | median | 0.07 | 0.05 | 0.02 | 0.02 | 0.04 | 0.05 | 0.02 | 0.02 | 0.02 |
| RSS | mean | 0.08 | 0.09 | 0.04 | 0.02 | 0.03 | 0.04 | 0.02 | 0.06 | 0.03 |
|  | SD | 0.07 | 0.07 | 0.03 | 0.01 | 0.02 | 0.03 | 0.01 | 0.11 | 0.02 |
|  | median | 0.05 | 0.07 | 0.04 | 0.01 | 0.02 | 0.03 | 0.02 | 0.03 | 0.02 |
| GA | mean | 0.04 | 0.03 | 0.02 | 0.01 | 0.02 | 0.02 | 0.02 | 0.01 | 0.01 |
|  | SD | 0.03 | 0.02 | 0.01 | 0.01 | 0.01 | 0.01 | 0.01 | 0.00 | 0.00 |
|  | median | 0.02 | 0.02 | 0.02 | 0.01 | 0.02 | 0.02 | 0.01 | 0.01 | 0.01 |
| AS | mean | 0.03 | 0.02 | 0.01 | 0.01 | 0.02 | 0.02 | 0.01 | 0.01 | 0.01 |
|  | SD | 0.04 | 0.01 | 0.00 | 0.01 | 0.01 | 0.01 | 0.00 | 0.00 | 0.00 |
|  | median | 0.02 | 0.02 | 0.01 | 0.01 | 0.01 | 0.02 | 0.01 | 0.01 | 0.01 |
| GO | mean | 0.05 | 0.05 | 0.03 | 0.02 | 0.03 | 0.02 | 0.02 | 0.02 | 0.01 |
|  | SD | 0.07 | 0.05 | 0.03 | 0.01 | 0.01 | 0.01 | 0.01 | 0.01 | 0.00 |
|  | median | 0.03 | 0.03 | 0.02 | 0.02 | 0.03 | 0.02 | 0.02 | 0.01 | 0.01 |
| AG | mean | 0.15 | 0.13 | 0.05 | 0.07 | 0.11 | 0.12 | 0.06 | 0.04 | 0.03 |
|  | SD | 0.15 | 0.10 | 0.04 | 0.07 | 0.11 | 0.10 | 0.05 | 0.03 | 0.02 |
|  | median | 0.11 | 0.10 | 0.04 | 0.04 | 0.08 | 0.09 | 0.04 | 0.03 | 0.02 |


**3.1.1    Aliphatic Carbonyls** ($C_nH_{2n}O$)
Relatively high mean mixing ratios of aliphatic carbonyls were observed over the Arabian Gulf; the highest being acetone (C3
carbonyl compound) at 4.50 ± 2.40 ppb (median: 3.77 ppb), followed by formaldehyde at 3.83 ± 2.55 ppb (median: 3.02 ppb),
acetaldehyde at 1.73 ± 1.61 ppb (median: 1.02 ppb) and MEK (C4 carbonyl compound) at 0.87 ± 0.71 ppb (median: 0.56 ppb).
The level of each aliphatic carbonyls over the Arabian Gulf was comparable to those previously reported for urban areas (Table
2), despite these measurements being taken at sea. As the Arabian Gulf is highly impacted by the oil and gas industry, we also
compared the measurements of the four aforementioned carbonyl compounds with those measured in the oil and gas region of the
Uinta Basin on land (Stoeckenius and McNally, 2014). Although the levels of three aliphatic carbonyls are higher in the Uinta
Basin (mean levels of 8 ppb, 4ppb and 2.8 ppb for acetone, acetaldehyde and MEK, respectively), formaldehyde was much lower
(1.9 ppb). The general distribution of the aliphatic carbonyls in the Uinta Basin is similar to the Arabian Gulf, with acetone levels
being twice as those of acetaldehyde. Koss et al. (2017) reported the max boundary layer enhancement of carbonyl compounds
(C2-C7) measured during an aircraft measurement above the most productive oil field in the United States (Permian Basin). Within
the boundary layer of the Permian Basin, C5-C7 aliphatic carbonyls had mixing ratios of 0.34 ppb, 0.08 ppb and 0.03 ppb; which



are of the same magnitude but lower than the levels measured over the Arabian Gulf for C5 (0.52 ± 0.48 ppb), C6 (0.19 ± 0.25ppb)
and C7 (0.05 ± 0.04 ppb) carbonyl compounds.
In contrast, aliphatic carbonyls had much lower average mixing ratios over the Arabian Sea and the Gulf of Aden especially for
C7-C9 carbonyls with mean mixing ratios below the detection limit for most of the time. During the summertime AQABA
campaign, the prevailing wind direction over the Arabian Sea was southwest (Figure S1). Four-day back trajectories indicate the
air was transported from the Arabian Sea (Northwestern Indian Ocean), passing East Africa coast, which brought relatively clean,
photochemically aged airmasses (Bourtsoukidis et al., 2019). The mean level of acetone over the Arabian Sea (0.43 ± 0.18 ppb,
median: 0.34 ppb) is close to the level measured in the marine boundary layer of Western Indian Ocean (0.49 ppb) (Warneke and
de Gouw, 2001) and comparable to other reported values from remote marine air measurement (see Table 2).  Acetaldehyde was
measured at relatively low mixing ratios over the Arabian Sea (median: 0.12 ppb), which is lower than the levels reported in most
ground-level marine influenced sites (Lewis et al., 2005;Read et al., 2012) but comparable to the value in Barrow Alaska (0.10 ±
0.20 ppb) (Hornbrook et al., 2016) and the values reported for Southern Indian Ocean (0.12 ± 0.04 ppb) (Colomb et al., 2009).
Over the Gulf of Aden, acetone and MEK had slightly higher mixing ratios than those over the Arabian Sea.
The Mediterranean Sea had somewhat higher levels of aliphatic carbonyls than the clean regions (the Arabian Sea and the Gulf of
Aden) but with acetone (above 2ppb) still dominating the distribution. The mean values of acetaldehyde, acetone and MEK are
comparable with the results from a rural site on the west coast of Cyprus (Derstroff et al., 2017). Larger aliphatic carbonyls (C6-
C9) were below the detection limit most of the time. The aliphatic carbonyls levels over the Gulf of Oman were higher than the
clean regions, while C1-C5 carbonyls were more variable over the Gulf of Oman compared to those over the Mediterranean Sea.
This is probably because the Gulf of Oman connects to the Arabian Gulf where intense oil and gas industrial activities are located.
Over the Gulf of Oman, polluted air from the nearby sources of the Arabian Gulf is occasionally mixed with the clean air from the
open sea (the Arabian Sea) under southeast wind conditions (Figure S1).
Another region where abundant aliphatic carbonyls were observed was Suez region. The air in this region was mainly influenced
by nearby cities and marine transportation (ship emissions within the Suez Chanel) (Bourtsoukidis et al., 2019;Pfannerstill et al.,
2019). However, the levels of acetaldehyde and MEK were much less compared to the levels reported from urban sites (see Table
2). Interestingly, the mean acetaldehyde mixing ratio (0.62 ± 0.59 ppb) over Suez was twice the level found over the Mediterranean
Sea, whilst the acetone level was only slightly higher. Besides the local-scale emission and photochemical production contribution
to the acetone over Suez, the longer lived acetone could be also transported from the Mediterranean Sea (where acetone was high).
Although the mean mixing ratios of aliphatic carbonyls over Suez were lower than those over the Arabian Gulf, larger variations
were observed.
Over the Red Sea, acetone was the most abundant aliphatic carbonyls followed by formaldehyde and acetaldehyde. The mixing
ratios of aliphatic C2-C4 carbonyls over the northern part of the Red Sea were similar to those levels measured in Thompson Farm
(a rural site in the US, Table 2). It is worth noticing that the levels of aliphatic carbonyls in the northern part of the Red Sea were
almost two times higher than the southern part of the Red Sea. According to the four-day back trajectories reported by
Bourtsoukidis et al. (2019), the measured air masses travelled to the northern part was from southern Europe and northeast Africa
while the southern part was more influenced by the air from the northen part Red Sea mixed with the air masses from desertic
areas of central Africa. Therefore, the higher carbonyl mixing ratios over the northern part Red Sea could be due to sources of
carbonyl precursors nearby and also the influence of aged air from over the Mediterranean Sea and polluted air from Suez region.



Table 2. Reported mixing ratios (ppb) of OVOCs

| Location | Time | Technique | Acetaldehyde | Acetone | MEK | HCHO | Literature |
|---|---|---|---|---|---|---|---|
| **Marine (Sea)** | | | | | | | |
| Mace Head, Ireland | Jul.-Sep. | GC-FID | 0.44 (0.12-2.12) | 0.50 (0.16-1.67) | n.r. | n.r. | (Lewis et al., 2005) |
| Appledore Island, USA | Jul.-Aug. | PTR-MS | 0.40 | 1.5 | 0.20 | n.r. | (White et al., 2008) |
| Western North Pacific Ocean | May | PTR-MS | n.r. | 0.20-0.70 | n.r. | n.r. | (Tanimoto et al., 2014) |
| Canadian Archipelago | Aug.-Sep. | PTR-MS | n.r. | 0.34[a] | n.r. | n.r. | (Sjostedt et al., 2012) |
| Western Indian Ocean | Feb.-Mar. | PTR-MS | n.r. | 0.49 | n.r. | n.r. | (Warneke and de Gouw, 2001) |
| Cape Verde Atmospheric Observatory | 2006-2011 | GC-FID | 0.43 (0.19-0.67) | 0.55 (0.23-0.91) | n.r. | n.r. | (Read et al., 2012) |
| Barrow Arctic, Alaska | Springtime 2009 | TOGA | 0.10 ± 0.20 | 0.90 ± 0.30 | 0.19 ± 0.05 | n.r. | (Hornbrook et al., 2016) |
| Southern Indian Ocean | Dec. 2004 | PTR-MS | 0.12 - 0.52 | 0.42 – 1.08 | n.r. | n.r. | (Colomb et al., 2009) |
| **Urban** | | | | | | | |
| Paris | Jan.- Feb. | PTR-MS/ GC-MS | 1.87 (0.92-4.49) | 1.05 (0.58-2.97) | n.r. | n.r. | (Dolgorouky et al., 2012) |
| Hong Kong | Sep.-Nov. | HPLC | 2.17[b] | n.r. | n.r. | 2.93[b] | (Guo et al., 2013) |
| Ahmedabad | Mar. | PTR-Tof-MS | 4.84 (2.74-9.86) | 5.63 (3.12-12.9) | n.r. | n.r. | (Sahu et al., 2017) |
| Beijing | Winter (clear day) | PTR-ToF-MS | 4.37 | 2.22 | 2.53 | 18.32 | (Sheng et al., 2018) |
| **Oil & Gas region** | | | | | | | |
| Uintah Basin, USA | 2013 | PTR-MS | 4.0 | 8.0 | 2.8 | 1.9 | (Stoeckenius and McNally, 2014) |
| **Rural** | | | | | | | |
| Thompson Farm, USA | Long-term (Summer) | PTR-MS | 0.54 (0.21-1.27) | 2.11 (0.98-4.08) | 0.22 (0.08-0.60) | n.r. | (Jordan et al., 2009) |
| Cyprian rural site | Jul.-Aug. | PTR-Tof-MS | 0.29[a] | 2.4[a] | 0.10[a] | | (Derstroff et al., 2017) |
| **Forest** | | | | | | | |
| Brazilian mixed tropical rainforest site | Feb.- Oct. | PTR-MS | n.r. | n.r. | 0.13[c] | n.r. | (Yáñez-Serrano et al., 2016) |

n.r.: not reported in the literature.
[a] Averaged value of reported values in the literature.
[b] converted from values in unit of ugm-3 in the literature
[c] daytime average









### 3.1.2 Unsaturated and aromatic carbonyls ($C_nH_{2n-2}O$), ($C_nH_{2n-8}O$)

The mixing ratios of unsaturated carbonyls were generally low with values below 30 ppt over the Mediterranean Sea and the clean regions (the Arabian Sea and the Gulf of Aden, 12 - 21 ppt). The Red Sea region and the Gulf of Oman had slightly higher levels (13 – 60 ppt). The highest values were again observed in the Arabian Gulf (25 – 115 ppt) followed by Suez (11 – 68 ppt). In terms of the mixing ratio distribution (Figure 2), the peak value was usually observed at C5 or C6 unsaturated carbonyls over most regions except for Suez where C4 carbonyl had the highest mixing ratio. Based on chemical formulas, unsaturated carbonyls can be either cyclic carbonyl compounds or carbonyls containing a carbon-carbon double bond. Therefore, the air chemistry could differ considerably depending on the compound assignment. A detailed analysis of the chemistry of the unsaturated carbonyls measured will be given in the following section 3.2.2.

Regional variability was also observed for aromatic carbonyls with highest levels observed over the Arabian Gulf and Suez, and much lower mixing ratios over the Arabian Sea, Mediterranean Sea and Gulf of Aden (Table 1). Several studies using PTR-MS have reported values for m/z 107.049 (C7 aromatic carbonyls) attributed to benzaldehyde (Brilli et al., 2014; Koss et al., 2017;Koss et al., 2018), m/z 121.065 (C8 aromatic carbonyls) attributed to tolualdehyde (Koss et al., 2018) or acetophenone (Brilli et al., 2014) and m/z 135.080 (C9 aromatic carbonyls) attributed to methyl acetophenone (Koss et al., 2018) or benzyl methyl ketone (Brilli et al., 2014) or 3,5-dimethylbenzaldehyde (Müller et al., 2012). Atmospheric aromatic carbonyls are produced via photochemical oxidation of aromatic hydrocarbons (Finlayson-Pitts and Pitts Jr, 1999;Wyche et al., 2009;Müller et al., 2012) and benzaldehyde was reported as having primary sources from biomass burning and anthropogenic emissions (Cabrera-Perez et al., 2016). Around the Arabian Peninsula, the level of aromatic carbonyls declined with increasing carbon number over most of the regions except in the Red Sea South where C8 carbonyls were slightly higher than C7 (Figure 2). Interestingly, only in the Suez region, were the C7 aromatic carbonyls more abundant than other aromatic carbonyls, whereby the mean value ($128 \pm 229$ ppt) was much higher than the median value (30 ppt), indicating strong primary sources of benzaldehyde in Suez. Otherwise, toluene was found to be more abundant over Suez with mean mixing ratios of $271 \pm 459$ ppt than over other regions (the mean over the Arabian Gulf: $130 \pm 160$ ppt) which would also lead to higher benzaldehyde as it is one of the OH-induced oxidation products of toluene via H-abstraction (Ji et al., 2017).

### 3.2 Chemistry of aliphatic carbonyls

#### 3.2.1 Importance of OH photochemistry

Aliphatic Carbonyls are a major fraction of all oxygenated volatile organic compounds (OVOCs) in the atmosphere. They can be directly emitted into the atmosphere; however, secondary production via photo oxidation of hydrocarbons is considered as the dominant atmospheric source. In order to better understand the contribution of hydrocarbon oxidation to aliphatic carbonyls in the AQABA region, we performed empirical calculations based on the measured precursor hydrocarbon levels to estimate secondary produced aliphatic carbonyls. Subsequently, we compared the calculated values with the measured levels of the aliphatic carbonyls. The calculations are based on the following assumptions: (1) the production and the sinks of aliphatic carbonyls are governed by OH radicals; (2) primary sources of aliphatic carbonyls are insignificant; (3) only methane and 11 other measured hydrocarbon species (Table S4) reported by (Bourtsoukidis et al., 2019) were considered in the calculation. The concentration of each aliphatic carbonyl can be calculated as follows:

$$[HC]_{initial} = [HC]/\exp(-k_{OH+HC}[OH]\Delta t) \qquad \text{Eq. (1)}$$

$$[Aliphatic\ carbonyls]_i = \sum(([HC]_{initial} \cdot (1 - \exp(-k_{OH+HC}[OH]\Delta t) \cdot Y) \cdot \exp(-k_{OH+Aliphatic\ carbonyls}[OH]\Delta t)) \qquad \text{Eq. (2)}$$



$[HC]_{initial}$ represents the initial mixing ratios of hydrocarbons from the source, which could be calculated using equation (1),
where $[HC]$ stands for the mean mixing ratios of measured hydrocarbons. The parameters $k_{OH+NMHC}$ (equation 1 and 2) and
$k_{OH+Aliphatic\ carbonyls}$ (equation 2) are the rate constants for the reaction between the OH radical and the corresponding
hydrocarbon and aliphatic carbonyl (see Table S3). For carbonyls with a carbon number larger than four, an average reaction rate
constant for the possible isomeric ketones was used since the exact structure is not known. Y is the yield of aliphatic carbonyls
produced in hydrocarbon oxidation, derived from the Master Chemical Mechanism, MCM v3.2 via website:
http://mcm.leeds.ac.uk/MCM (last accessed on Jan-15, 2020) (Jenkin et al., 1997;Saunders et al., 2003). The yields (Table S4)
were determined assuming that the oxidation is dominated by OH chemistry and that alkylperoxy radicals (RO₂) mainly react with
NO. The OH exposure term $[OH]\Delta t$ in both equations was derived from a method based on hydrocarbon ratios (Roberts et al.,
1984;de Gouw et al., 2005;Yuan et al., 2012), and was calculated using the following equation:
$$[OH]\Delta t = \frac{1}{k_X - k_Y} \cdot \left( In \left. \frac{[X]}{[Y]} \right|_{t=0} - In \frac{[X]}{[Y]} \right) ,$$ Eq. (3)
where X and Y refer to two hydrocarbon compounds with different rates of reaction with the OH radical (k). For this study, we
chose toluene ($k_{OH+toluene}$: 5.63E-12 cm³ molecule⁻¹s⁻¹ ) and benzene ($k_{OH+benzene}$: 1.22E-12 cm³ molecule⁻¹s⁻¹) (Atkinson and Arey,
2003), because both compounds were measured by PTR-ToF-MS at high frequency and these values showed a good agreement
with values measured by GC-FID (Figure S2). The approach detailed by Yuan et al. (2012) was applied to determine the initial
emission ratio $\left. \frac{[X]}{[Y]} \right|_{t=0}$ in each area: MS (1.46), S (2.50), RSN (1.25), RSS (2.50), GA (1.96), AS (1.25), GO (2.50) and AG (1.75).
The corresponding correlation plots of toluene and benzene for each region can be found in Figure S3. An average value of OH
exposure in each area was applied in equation (2) to derive the predicted mixing ratio of each aliphatic carbonyl. There are
limitations and uncertainties in using the hydrocarbon ratios to obtain the photochemical age $\Delta t$ (OH exposure in this study),
especially in remote area lacking hydrocarbon (benzene and toluene in this case) sources, which would introduce uncertainty to
the initial emission ratio determination (Yuan et al., 2012;Lewis et al., 2005). Nevertheless, it can still provide insights on
photochemical processing (Parrish et al., 2007;McKeen et al., 1996), which will be discussed afterwards.
Figure 3 shows the calculated mixing ratios of the aliphatic carbonyls and the unattributed fraction compared with the mean values
of measurements in each area. The calculated carbonyl mixing ratios are sub-divided in the plot to show the fraction stemming
from a parent hydrocarbon with the same carbon number, and the fraction resulting from fragmentation of a hydrocarbon with a
higher carbon number. In general, the direct oxidation fraction varied from area to area for C1 to C3 carbonyls (formaldehyde,
acetaldehyde and acetone). For C4 and C5 carbonyls, the mixing ratios could be fully attributed to the direct oxidation fraction
over the Arabian Gulf and Suez (polluted region with high loadings), while in other regions, direct oxidation fraction could only
partially explain the measured mixing ratios. For C7 and C8 carbonyls, the oxidation fraction generally explained < 20% of
measured mixing ratios. Notably, for acetone, the oxidation (mainly from propane oxidation) contributed much more of the direct
oxidation fraction over the Arabian Gulf (53%) and Suez (35%) than those in other regions (~10%), indicating OH radical initiated
hydrocarbon oxidation (and subsequent secondary photochemical processes) play a more important role in polluted regions than
in other less polluted regions. This is also consistent with the results that showed C4 and C5 carbonyls mixing ratios could be fully
attributed by the direct oxidation fractions. The high-unattributed levels of acetone especially over the remote areas are possibly
due to transportation as its long lifetime from 15 days up to more than two months in the boundary layer (Singh et al., 1994)
compared to the other carbonyl compounds.



The distribution for acetaldehyde was markedly different to that of acetone. The measured mixing ratios were mainly determined
by the unattributed fraction, even over the Arabian Gulf and Suez, where the oxidation fraction only accounts for < 5%. Because
the OH rate coefficient for acetaldehyde (at 298 K) is two orders of magnitude larger than that for acetone, long-distance transport
of short-lived acetaldehyde was limited. Although oxidation from unconsidered hydrocarbons could be a reason, the much larger
unattributed fraction compared with that of acetone indicates that OH radical driven oxidation of the main hydrocarbons present
cannot explain the measured acetaldehyde level in this region. This points to the existence of other production pathways (primary
and/or secondary) of acetaldehyde. Therefore, we compared the measured acetaldehyde with the results from a complex
atmospheric chemistry model (EMAC) which includes transport and known sources to further investigate the discrepancy. This
discussion is given in section 3.4 and 3.5.
Formaldehyde had a more than 50% contribution from hydrocarbon oxidation over the Red Sea and Gulf of Oman but less than
40% over the Arabian Gulf. Over the Suez region, calculated formaldehyde level was even 24% higher than the measured mean
mixing ratio. The contribution from other hydrocarbons is more significant than from methane oxidation over the Arabian Gulf,
Suez and Gulf of Oman. Atmospheric formaldehyde has been reported as having various primary and secondary sources and,
unlike the other carbonyls, photolysis is an important sink (Carlier et al., 1986). Neither the primary sources nor the photolysis
were considered in the calculation. Therefore, different unattributed fractions in the Arabian Gulf and Suez suggest the existence
of different formaldehyde formation pathways. Similar to acetaldehyde, due to the short lifetime, the unattributed fraction indicates
unconsidered sources and formation pathways.

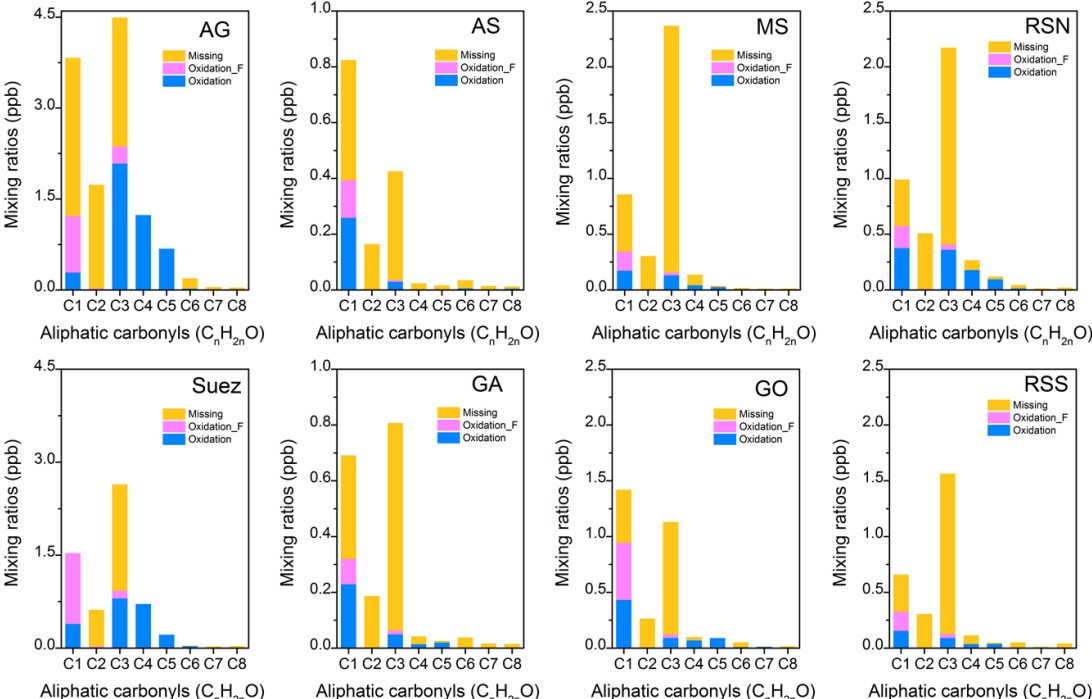


Figure 3. Measured and calculated aliphatic carbonyls as mentioned in the text for different regions during the AQABA campaign:
Oxidation (in blue) represents aliphatic carbonyls produced from alkane oxidation with the same carbon number; Oxidation_F
represents aliphatic carbonyls produced from other hydrocarbon oxidation; Missing (in yellow) is the unattributed fraction
compared with the mean values of measurements.





### 3.2.2 Case studies: the Arabian Gulf and Suez region

The primary emission sources in the Arabian Gulf and Suez regions are quite different. While the Arabian Gulf is dominated by oil and gas operations, Suez is more influenced by ship emissions and urban areas (Bourtsoukidis et al., 2019). Carbonyl compounds were most abundant in these two areas. As mentioned before, photochemical oxidation contributed a large fraction to acetone and the larger aliphatic carbonyls over the Arabian Gulf and Suez areas, but could not explain the high level of acetaldehyde measured in both regions. For further insight, we focused on a time series of selected trace-gases along with the correlations among them to better identify the sources of the major aliphatic carbonyls.

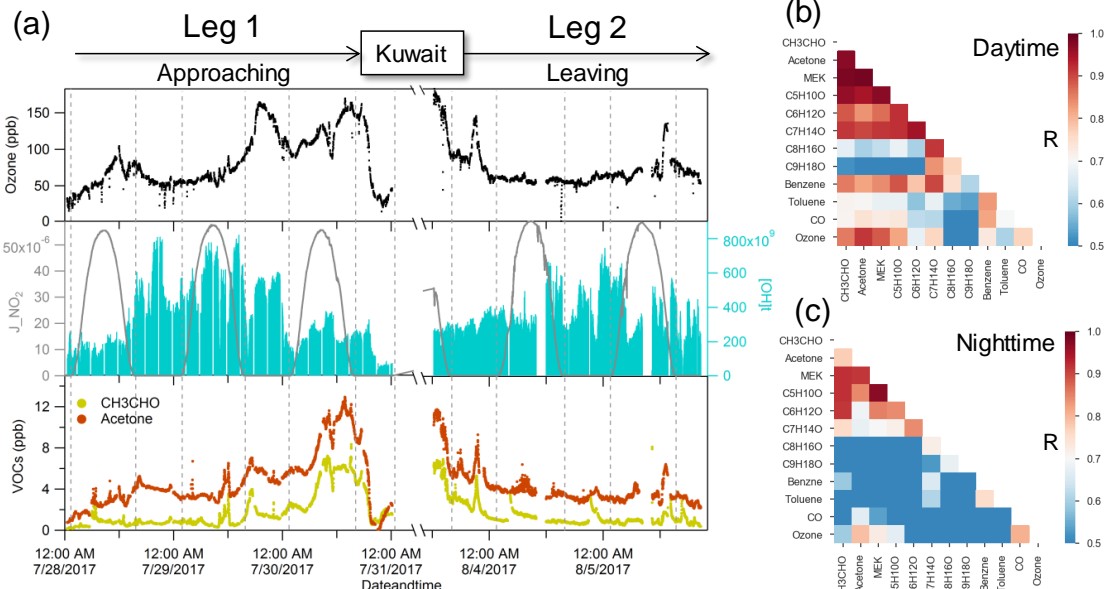

Figure 4. Case study of the Arabian Gulf. (a) Time series of selected species measured over the Arabian Gulf; (b) daytime correlation heat map of selected species; (c) nighttime correlation heat map of selected species.

Figure 4(a) shows the time series of acetaldehyde and acetone over the Arabian Gulf along with OH exposure ($[OH]t$) and ozone. We further separated the data into daytime and nighttime and calculated correlations among the carbonyls and other selected species (see Fig. 4b and c). Aliphatic carbonyls were well correlated with each other during the daytime and ozone had a generally good correlation with C2-C7 carbonyls (r > 0.7) during the daytime but a much lower correlation during the night, indicating ozone and carbonyls were co-produced via photochemical oxidation. This further emphasizes the importance of local photochemical production of aliphatic carbonyls over the Arabian Gulf, as suggested in previous section 3.2.1. Meanwhile, as shown in Figure 4 (a), the calculated OH exposure was high during the first night in leg 1, where an elevation of acetone mixing ratio was observed while the mixing ratio of acetaldehyde remained relatively constant. With limited OH radical abundance during the nighttime, the increased OH exposure indicates that the air reaching the ship was photochemically processed (aged). Therefore, the increase of acetone was mainly from long-distance transport as acetone has a much longer atmospheric lifetime than acetaldehyde. As the ship approached Kuwait, the calculated OH exposure was low (starting from 7/30/2017, 12:00 am UTC), which is an indicator of nearby emission sources. The lifetime of the OH radical derived from the measured OH reactivity also decreased from ~ 0.1 s to ~ 0.04 s during the same period (Pfannerstill et al., 2019). Oil fields and associated refineries are densely distributed in the northwest of the Arabian Gulf region (United States Central Intelligence Agency). The air reaching the ship when mixing ratios of acetone and





acetaldehyde were highest was mainly from the Northwest (Iraq oil field region) according to the back trajectories (Bourtsoukidis
et al., 2019). This suggests that the air masses encountered in Northwest Arabian Gulf were a combination of fresh emissions from
nearby sources and photochemically processed air transported from elsewhere. During the second leg, relatively low mixing ratios
were identified in the same region (Northwest Arabian Gulf), which was mainly due to a greater influence of air masses originating
from less populated desert regions of Northeast Iran (Bourtsoukidis et al., 2019) with much less influence from the oil field
emissions, meaning less precursors were available for carbonyl production. Several plumes (extending over 2-3 hours) of elevated
carbonyls with increased ozone were observed during the nighttime for both legs (Fig. 4a), indicating transport of highly polluted
air.

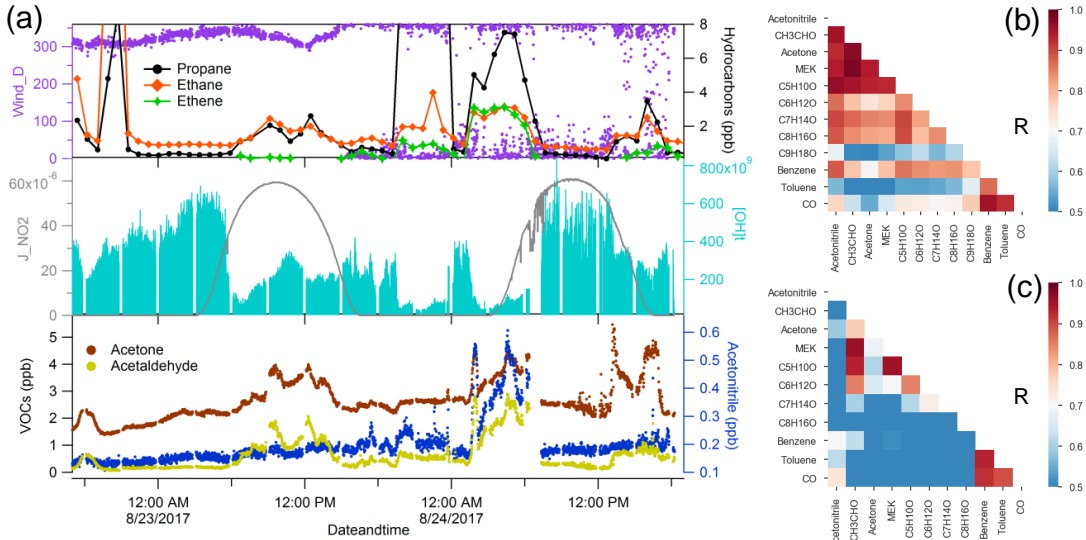


Figure 5. Case study of Suez. (a) Time series of selected species measured over Suez; (b) correlation heat map of selected species
during biomass burning plume (UTC 01:00 -06:00 August 24th 2017); (c) correlation heat map of selected species without the
period of biomass burning plume.
For the Suez region (Gulf of Suez and Suez Canal), data were only available for the second leg. A significant increase of acetonitrile
(over 400 ppt) was observed just before entering the Great Bitter Lake (see Figure 5a), indicating an increasing influence of biomass
burning on the air composition (Lobert et al., 1990). Carbonyl compounds are important primary emissions in fresh biomass
burning plumes (Holzinger et al., 1999;Schauer et al., 2001;Holzinger et al., 2001;Koss et al., 2018) as well as being formed as
secondary products in more aged plumes (Holzinger et al., 2005). We further investigated the correlation coefficient among
carbonyls during the biomass burning plume (Figure 5b) in Suez. Carbonyls had a high correlation with acetonitrile, benzene and
among themselves, particularly for smaller carbonyls (acetaldehyde, C3-C5 carbonyls). The biomass burning emissions were
probably transported by on the prevailing northerly wind above Northeast Egypt (southern side of Suez Canal) where crop residues
especially rice straw is often directly burned in the open fields (Abdelhady et al., 2014;Said et al., 2013;Youssef et al., 2009).
Besides the direct biomass burning emission, the high mixing ratios and the good correlations of carbonyls could also have resulted
from other sources as hydrocarbons (alkanes, alkenes and aromatics) which were elevated at the same time. Similar to conditions
identified over the Arabian Gulf, elevated OH exposure accompanied with increasing acetone mixing ratio was observed during
the first night over the Gulf of Suez, indicating aged air mass transportation. The OH exposure was then significantly lower during
the daytime, when mixing ratios of carbonyls and alkanes increased as well. This indicates the presence of emission sources nearby.



Oil refineries located in the costal side of Suez and oil tank terminals located in the northern part of the Gulf of Suez are likely
sources.

### 3.3  Air chemistry of unsaturated carbonyls

Unsaturated carbonyls measured by PTR-MS have been only rarely reported in the atmosphere with the exception of methyl vinyl
ketone and methacrolein (C4 carbonyls) which are frequently reported as the oxidation products of isoprene (Williams et al., 2001;
Fan and Zhang, 2004;Wennberg et al., 2018). According to the GC-FID measurement, isoprene was below the detection limit for
most of the time during the AQABA cruise with the highest values observed in Suez (10 - 350 ppt). This shows that the AQABA
campaign was little influenced by either terrestrial or marine isoprene emissions. However, we observed unexpected high levels
on mass 69.070, which is usually interpreted as isoprene for PTR-MS measurements. Significant enhancements were even
identified while sampling our own ship exhaust (in PTR-MS but not GC-FID), suggesting the presence of an anthropogenic
interference at that mass under these extremely polluted conditions. Several studies have reported possible fragmentations of cyclic
alkanes giving mass (m/z) 69.070. These include: a laboratory study on gasoline hydrocarbon measurements by PTR-MS
(Gueneron et al., 2015), a GC-PTR-MS study of an oil spill site combined with analysis of crude oil samples (Yuan et al., 2014)
and an inter-comparison of PTR-MS and GC in an O&G industrial site (Warneke et al., 2014). From those studies, other
fragmentations from C5-C9 cycloalkanes including m/z 43, m/z 57, m/z 83, m/z 111 and m/z 125 were identified together with
m/z 69. Cyclic alkanes were directly measured in oil and gas fields (Simpson et al., 2010;Gilman et al., 2013; Li et al., 2017;Aklilu
et al., 2018), vehicle exhaust (Gentner et al., 2012;Erickson et al., 2014), vessel exhaust (Xiao et al., 2018), accounting for a non-
negligible amount of the total VOCs mass depending on the fuel type. Koss et al. (2017) reported enhancement of cyclic alkane
fragment signals and increased levels of unsaturated carbonyls measured by PTR-ToF-MS over O&G region in the US. The
unsaturated carbonyls (C5-C9) were assigned as oxidation products of cycloalkanes. Therefore, we examined the correlations
between m/z 69.070 and other cycloalkane fragments over the Arabian Gulf and Suez, where anthropogenic primary emissions
were significant. As shown in Figure 6, m/z 83 was the most abundant fragment and it correlated better with m/z 69 than the other
two masses, strongly supporting the presence of C6 cycloalkanes (methylcyclopentane and cyclohexane). The other two masses
are distributed in two or three clusters, suggesting compositions of different cycloalkanes. M/z 43 and m/z 57 (fragments of C5
cycloalkanes) had lower correlations with other fragments (not shown in the graph) as they are also fragments of other higher
hydrocarbons. Thereby we could assign those unsaturated carbonyls as photochemical oxidation products (i.e. cyclic ketones or
aldehydes) from their precursor cycloalkanes.

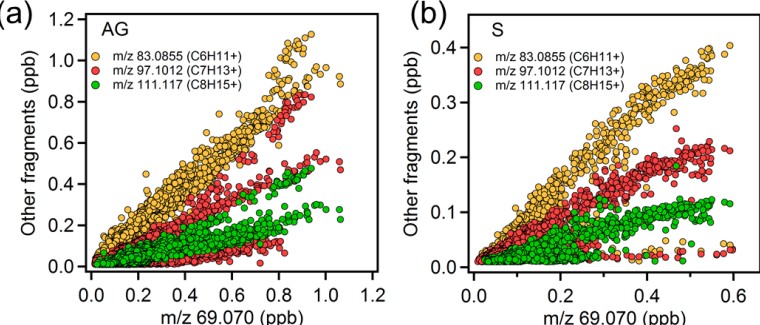

Figure 6. Scatter plots of m/z 69.070 and other cycloalkane fragment masses over the (a) Arabian Gulf and (b) Suez region.



As shown in Figure 2 and Table 1, C6 unsaturated carbonyls displayed higher mixing ratios than any other unsaturated carbonyls over the Arabian Gulf while C5 unsaturated carbonyl was slightly higher than C6 in Suez. Bourtsoukidis et al. (2019) derived enhancement ratio slopes from pentane isomers and established that the Arabian Gulf is dominated by oil and gas operations and that Suez is more influenced by ship emissions. Therefore, as the Arabian Gulf had much more active O&G activities than Suez, our findings agree with Koss et al. (2017) who showed that C6 unsaturated carbonyls should be more abundant than C5 carbonyls since more precursors for C6 unsaturated carbonyls are emitted from active oil fields. It is worth mentioning that in Figure 6 (b) one cluster at the bottom showed m/z 69.070 had no correlation with other three masses. Those points correspond to the time when the GC measured significant elevated isoprene while passing through the narrow Suez Canal where some vegetation (e.g. palms and some agriculture) was present close to shore, meaning m/z 69.070 during this period was isoprene. At the same time, m/z 71.049 (C4 unsaturated carbonyl) increased from 20 ppt to 220 ppt. Isoprene oxidation products (MVK and methacrolein) were probably the major contribution to the C4 unsaturated carbonyls in this period. This also explains why C4 carbonyl dominated the distribution of unsaturated carbonyls over Suez.

In the other regions (especially more remote areas), the cyclic alkane fragmentation masses had much lower abundance, leading to much less unsaturated carbonyls due to lack of precursors. Meanwhile, m/z 69.070 ($C_5H_8H^+$), m/z 83.086 ($C_6H_{10}H^+$) and m/z 97.101 ($C_7H_{12}H^+$) could also be fragmentations from corresponding aldehydes losing one water molecule as mentioned in section 2.3.3. Missing information of the chemical structure of unsaturated carbonyls and knowledge of their precursors, preclude detailed investigation of the sources of large unsaturated carbonyls in these areas.

### 3.4 Model comparison of acetaldehyde, acetone and MEK

We compared our measurement results of acetaldehyde, acetone and MEK to those predicted by the global model "EMAC" (ECHAM5/MESSy2 for Atmospheric Chemistry). The model considers direct emissions (such as anthropogenic, biogenic, biomass burning etc.), atmospheric transport and mixing, photochemical production of carbonyls (by OH, $O_3$ and $NO_3$), and physical and chemical removal processes. From the results shown in Figure 7, the model predicted acetone much better than acetaldehyde and MEK. In general, the model broadly captured the major features identified during the campaign such as much higher levels of carbonyls mixing ratios over the Arabian Gulf and Suez and relatively low levels over the Arabian Sea. The mean measurements-to-model ratios indicated that acetone was overestimated by a factor within 1.5 over the Arabian Sea, Gulf of Aden and Gulf of Oman, and underestimated by a factor within 2.5 over the other regions. In contrast, the model underestimated MEK within a factor of 4 over most of the regions except for the Gulf of Oman where MEK was overestimated (median values were taken here as the mean values substantially deviated from the medians over Suez, Gulf of Oman and Arabian Gulf). The model underestimation was most significant for acetaldehyde, which is underpredicted by a factor (median values) of more than 6 over the Red Sea North, ~ 4 over the Arabian Sea and Arabian Gulf and between 1 and 4 over other regions. A strong natural non-methane hydrocarbon source from deep water in the Northern Red Sea was implemented in the model (Bourtsoukidis et al., 2020). Although the model representation of acetaldehyde and other carbonyls was clearly improved after including the deep water source of ethane and propane (Figure S4), the underestimation of acetaldehyde was still significant over the Red Sea North as shown in Figure 7(a), indicating further missing sources. For acetaldehyde and MEK, the discrepancy was also significant over the Arabian Sea where acetone was in contrast, overestimated. Since acetaldehyde had the biggest bias from the model prediction both with our simple empirical calculation (section 3.2.1) and the global model, we further investigate the possible missing sources of acetaldehyde.



478

Figure 7. Measurement to model ratios (left) and time series (right) of measurements (in black) and model simulation (in red) of
(a) acetaldehyde; (b) acetone; (c) MEK in each area. In each box plot, the box represents 25% to 75% of the data set with central
line and square indicating the median value and the mean value respectively. The whiskers show data from 10% to 90%. The red
dashed lines represent the 1:1 ratio.

### 3.5  Missing sources of acetaldehyde

In this section we investigate the following processes as potential sources of acetaldehyde: (1) production as an inlet artifact, (2)
oceanic emission of acetaldehyde, (3) anthropogenic primary sources, (4) biomass burning sources, and (5) other possible
secondary formation pathways.

### 3.5.1    Inlet artifact

Northway et al. (2004) and Apel et al. (2008) reported that heterogeneous reactions of unsaturated organic species with ozone on
the wall of the Teflon inlet can cause artifacts signal of acetaldehyde but not to acetone. During AQABA, the highest and the most





variable ozone mixing ratios were observed during the campaign over the Arabian Gulf (mean: $80 \pm 34$ ppb) and the Red Sea North
($66 \pm 12$ ppb), where a modest correlation was found between acetaldehyde and ozone over the Arabian Gulf ($r^2=0.54$) and no
significant correlation over the Red Sea North ($r^2=0.40$). However larger correlation coefficients were identified between ozone
and other carbonyls over the Arabian Gulf (see Figure S5), which suggests that the correlation was due to atmospheric
photochemical production rather than artifacts. Moreover, acetaldehyde was found to have a much worse correlation with ozone
during the nighttime compared to the correlation during the daytime over the Arabian Gulf (Figure 4b and c), which also indicates
that inlet generation of acetaldehyde was insignificant. Over other regions, especially the remote area (the Arabian Sea and Gulf
of Aden), ozone was relatively constant and low, with poor correlation with acetaldehyde mixing ratios. Although we cannot
completely exclude the possible existence of artifacts, the interference is likely to be insignificant in this dataset.

### 3.5.2    Oceanic emission

A bias between measured acetaldehyde and global model simulations has been observed in previous studies conducted in the
remote troposphere (Singh et al., 2003; Singh, 2004;Wang et al., 2019) and in the marine boundary layer (Read et al., 2012). The
aforementioned studies emphasized the potential importance of the sea water acting as a source of acetaldehyde emission via air-
sea exchange. No significant correlation was found between acetaldehyde and DMS, a marker of marine biogenic emission which
is produced by phytoplankton in seawater (Bates et al., 1992) (see Figure S6). This indicates that the source of acetaldehyde was
probably not from direct biogenic production, which has been reported by Mungall et al. (2017). More likely, acetaldehyde and
other small carbonyl compounds can be formed in the sea especially in the surface microlayer (SML) via photodegradation of
colored dissolved organic matter (CDOM) (Kieber et al., 1990;Zhou and Mopper, 1997;Ciuraru et al., 2015). Zhou and Mopper
(1997) calculated the exchange direction of small carbonyls based on measurement results and identified that the net flux of
acetaldehyde was from sea to the air whereas formaldehyde was taken up by the sea. Sinha et al. (2007) characterized air-sea flux
of several VOCs in a mesocosm experiment and found that acetaldehyde emissions were in close correlation with light intensity
(r=0.7). By using a 3-D model, Millet et al. (2010) estimated the net oceanic emission of acetaldehyde to be as high as 57 Tg a$^{-1}$
(in a global total budget: 213 Tg a$^{-1}$), being the second largest global source. A similar approach was applied in a recent study done
by Wang et al. (2019), reporting the upper limit of the net ocean emission of acetaldehyde to be 34 Tg a$^{-1}$. To our knowledge, there
is no clear experimental evidence showing the ocean to be a sink for acetaldehyde.
In order to test the importance of the oceanic emission of acetaldehyde, we implemented this source in EMAC model. The measured
sea water concentration of acetaldehyde was not available for the water area around the Arabian Peninsula. Wang et al. (2019)
estimated the global average acetaldehyde surface seawater concentrations of the ocean mixed layer using a satellite-based
approach similar to Millet et al. (2010), where the model estimation agreed well with limited reported measurements. From the
Wang et al. (2019) results, the averaged seawater concentration of acetaldehyde around Arabian Peninsula was generally much
higher from June to August. As the photodegradation of CDOM is highly dependent on sunlight, the air-sea submodel (Pozzer et
al., 2006) was augmented to include throughout the campaign a scaled acetaldehyde seawater concentration in the range of 0 ~ 50
nM according to the solar radiation (Figure S7). With this approach, the average of acetaldehyde seawater concentration estimated
by the model is 13.4 nM, a reasonable level compared to predicted level by Wang et al. (2019).
After adding the oceanic source of acetaldehyde, the model estimation was significantly improved (Figure 8). As the oceanic source
in the model is scaled according to the solar radiation, the measurement-to-model ratios were more strongly reduced during the
day compared to the night. With oceanic emission included, the model underestimation was less significant, within a factor of 3
during the day and 4 during the night over the Mediterranean Sea, Red Sea and Gulf of Aden. The most significant improvement
was identified over the Red Sea North. As shown in Figure 9, the model had much better agreement with the measurement after




adding the oceanic source. The scatter plots for other regions can be found in Figure S8. Over the Arabian Sea, the model
significantly overestimated acetaldehyde mixing ratios, indicating the input sea water concentration of acetaldehyde might be too
high. The SML layer starts to be effectively destroyed by the wave breaking when the wind speed exceeds than 8 m s$^{-1}$ (Gantt et
al., 2011). As the average wind speed over the Arabian Sea was the highest among the cruised areas (8.1 ± 2.4 m s$^{-1}$, Figure S1),
less contribution from the CDOM photo degradation to acetaldehyde in the surface sea water would be expected. For the Suez
region, due to the limited model resolution (1.1° × 1.1°), little sea water was identified in the model, leading to negligible influence
from the oceanic source.
Model underestimation of acetaldehyde especially over the Suez, Red Sea and Arabian Gulf is also likely to be related to the coarse
model resolution (~ 1.1° × 1.1°) (Fischer et al., 2015). Where model grid points contain areas of land the higher and more variable
terrestrial boundary layer height impacts the model prediction whereas the measurements may only by influenced by a shallower
and more stable marine boundary layer.

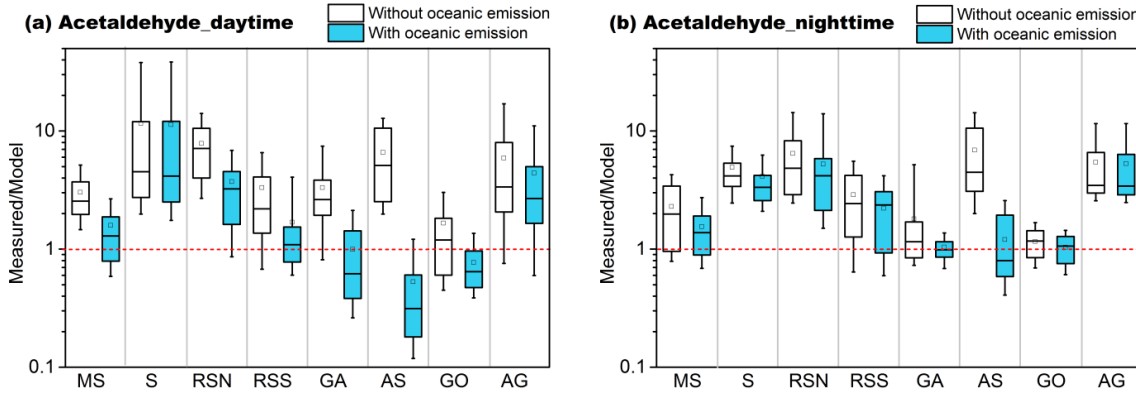


Figure 8. Acetaldehyde measurement to model ratios without the oceanic source (white boxes) and with the oceanic source (blue
boxes) in the model during (a) daytime and (b) nighttime in different regions. The boxes represent 25% to 75% of the data set
with the central line and square indicating the median and mean values, respectively. The whiskers show data from 10% to 90%.
The red dashed lines represent the 1:1 ratio.

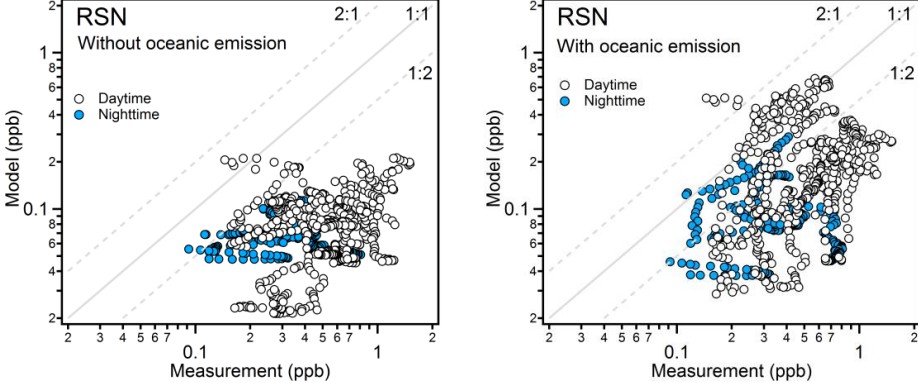


Figure 9. Observed and simulated mixing ratios of acetaldehyde over the Red Sea North without oceanic emission (left) and with
oceanic emission (right). The data points are separated into day- and nighttime according to solar radiation.





### 3.5.3 Anthropogenic primary sources

Over the Arabian Gulf and Suez, the intensive photochemical production of carbonyls is apparent. Therefore, an underestimation of the precursor hydrocarbons especially those large alkanes, alkenes and cyclic hydrocarbons which were not measured (> C12) or included in the model (> C5) could be a reason for the model underestimation of acetaldehyde and other carbonyls. Bourtsoukidis et al. (2020) compared measured hydrocarbons (ethane, propane, ethene etc.) with the results from model simulations (the same model used in this study) and periodically found significant model underestimations in both regions. This indicates that not all sources were present in the model's emission inventory. As mentioned in the previous case studies, high ozone mixing ratios were observed over the Arabian Gulf and Suez. With large amounts of alkenes present in those regions, which the model occasionally underestimated, the nighttime ozonolysis of alkenes could be another important source for acetaldehyde, formaldehyde and other carbonyls (Atkinson et al., 1995;Altshuller, 1993). Acetaldehyde, an oxygenated VOC, is not generally considered as an important primary emission from oil and gas field but instead a photochemical product of hydrocarbon oxidation (Yuan et al., 2014;Koss et al., 2015;Koss et al., 2017). In contrast, primary sources of formaldehyde from oil and gas production processes including both combustion and non-combustion process have been ascertained (Vaught, 1991).  Le Baron and Stoeckenius (2015) concluded in their report of the Uinta Basin winter ozone study that besides formaldehyde, the other carbonyls were poorly understood in terms of their primary sources. Acetaldehyde and other carbonyls (aldehydes and ketones) have been reported as primary emissions from fossil fuel combustion including ship emissions (Reda et al., 2014;Xiao et al., 2018;Huang et al., 2018) and vehicle emissions (Nogueira et al., 2014;Erickson et al., 2014;Dong et al., 2014). Therefore, the active petroleum industry located in the Arabian Gulf and intensive marine transportation in Suez are likely primary sources of acetaldehyde and other carbonyls which were not well constrained in the model. The Suez region, where the largest acetaldehyde discrepancy was identified, had a significant influence from biomass burning (see section 3.2.2). Biomass burning emissions are notoriously difficult to model as they are highly variable both in time and space. In this study, the model failed to reproduce the acetonitrile level with a range of only 40-50 ppt rather than 100-550 ppt measured over Suez. Thus, besides the possibility of seawater emission from the Gulf of Suez and the Suez Canal, the underestimated biomass burning source in the model over Suez, will lead to an underestimation of acetaldehyde as well as other carbonyl compounds in this region.

### 3.5.4 Other possible secondary formation pathways

Although the model estimation was generally improved with the addition of an oceanic source, the model to measured ratios still varied over a wide range. As mentioned above, photodegradation of CDOM on the surface of seawater is a known source for acetaldehyde although some studies focusing on real sea water samples did not observe clear diel cycles of seawater acetaldehyde (Beale et al., 2013;Yang et al., 2014). Fast microbial oxidation could be a reason (Dixon et al., 2013) while other non-light driven sources of acetaldehyde could be an alternative explanation. In a recent study, Zhou et al. (2014) reported enhanced gas-phase carbonyl compounds including acetaldehyde during a laboratory experiment of ozone reacting with SML samples, indicating acetaldehyde could also be produced under non-light driven heterogeneous oxidation. Wang et al. (2019) ventured a hypothetical source that organic aerosol can be an extra source for unattributed acetaldehyde in the free troposphere through light-driven production and ozonolysis. However, since the yield of acetaldehyde from such reactions is unknown, large uncertainties remain. Previous studies have shown that the organic matter fraction was highest in smaller sea spray aerosols and that the aerosols contain both saturated and unsaturated fatty acids originating from the seawater surface (i.e. SML) (Mochida et al., 2002;Cochran et al., 2016). Thus, for the AQABA campaign, both photodegradation and heterogeneous oxidation could occur on the surface of sea spray and pollution associated aerosols, even over remote open ocean therefore being an extra source of acetaldehyde and other carbonyl compounds. Another acetaldehyde formation pathway reported is gas-phase photolysis of pyruvic acid (Eger et al.,



2019b;Reed Harris et al., 2016), a compound mainly of biogenic origin. Pyruvic acid has been also observed in seawater (Kieber
and Mopper, 1987;Zhou and Mopper, 1997;Tedetti et al., 2006), although acetaldehyde was not the major product of aqueous-
phase photolysis of pyruvic acid (Griffith et al., 2013). Zhou and Mopper (1997) pointed out that the net exchange direction for
pyruvic acid is expected to be from the air to the sea due to its high partition coefficient (high solubility). Therefore, only low
levels of pyruvic acid would be expected in the remote marine boundary layer. Pyruvic acid was measured by Jardine et al. (2010)
using a PTR-MS at m/z 89 in a forested environment. For the AQABA PTR-ToF-MS data set, enhanced signals were observed at
m/z 89.024 with the mean mixing ratio of $58 \pm 34$ ppt (the box plot can be found in Figure S9), which is much more abundant than
reported pyruvic acid levels measured above Atlantic Ocean ($1.1 \pm 1.0$ ppt) (Baboukas et al., 2000). This might be due to the
uncertainty associated with the theoretical methods of quantification used here or the presence of isomeric compounds on that
mass, since pyruvic acid was not calibrated with the standard. As the air-sea exchange of pyruvic acid is limited, low levels of
pyruvic acid were expected. Even if we fully assign the m/z 89.024 to pyruvic acid, the contribution to acetaldehyde via photolysis
of pyruvic acid is negligible compared other sources. Therefore, we conclude that the contribution from the photolysis of pyruvic
acid is not an important source for the unattributed acetaldehyde during the AQABA campaign.

## 4    Summary and Conclusion

Observations of carbonyl compounds around the Arabian Peninsula were investigated in terms of mixing ratios abundance over
different areas. Aliphatic carbonyl compounds were generally more abundant than the unsaturated and aromatic carbonyl
compounds, and were dominated by low-molecular-weight compounds (carbon number less than five). Aliphatic carbonyl
compounds were found at the highest mixing ratios over the Arabian Gulf followed by the Suez region, while the lowest mixing
ratios were observed over the Arabian Sea and the Gulf of Aden. Over the Mediterranean Sea, aliphatic carbonyls were low except
for acetone that was much higher compared to the levels observed over clean remote areas (i.e. Arabian Sea). The atmospheric
composition over the Red Sea showed obvious differences between the northern and the southern part, with higher mixing ratios
in the north. Similar region-dependent distributions were observed for unsaturated and aromatic carbonyls. Generally, the mixing
ratios of aromatic carbonyl compounds decreased as the carbon number increased. Particularly over the Suez region, benzaldehyde
(C7 aromatic carbonyls) was much more abundant than other aromatic carbonyls, indicating direct sources as well as abundant
oxidation precursors. For unsaturated carbonyl compounds, C5 and C6 carbonyl compounds dominated the mixing ratio
distribution, while the air chemistry highly depends on the chemical structure assignment of those masses.
To better understand the air chemistry of aliphatic carbonyl compounds over different regions, we used an empirical method to
calculate the levels of carbonyl compounds resulting from OH oxidation of precursor hydrocarbon species. The results indicate
that mixing ratios of formaldehyde and C3-C8 carbonyl compounds could, to a large part, be explained by OH initiated
photooxidation in each region, especially over the Arabian Gulf and Suez region. This result indicates that photooxidation is a
dominant production pathway for formaldehyde and C3-C8 aliphatic carbonyl compounds in these two regions. However,
acetaldehyde from hydrocarbon precursors was not sufficient to explain the high mixing ratios observed, indicating the existence
of other sources and/or formation pathways. Further case studies showed that the carbonyl compounds produced via photooxidation
were highly correlated to the high ozone levels during daytime over the Arabian Gulf while the air chemistry in Suez region was
strongly influenced by regional biomass burning. Due to the unexpectedly high loading of m/z 69 (usually assigned as isoprene)
observed in highly polluted regions, we further identified the correlations between m/z 69 and other fragmentation masses of
cycloalkanes according to previous studies conducted in oil and gas regions (Warneke et al., 2014;Yuan et al., 2014;Koss et al.,
2017). The high correlations among fragments implied the existence of cycloalkanes in the polluted regions, which could be further
oxidized to unsaturated carbonyl compounds (cyclic ketones or aldehydes).



As acetaldehyde was identified as having important additional sources, we further compared the measurements of major carbonyl
species (acetaldehyde, acetone and MEK) with a comprehensive global atmospheric chemistry model (EMAC). Acetaldehyde was
found to have the highest discrepancy between the observations and model simulations, with the simulated values to be lower up
to a factor of 10. By adding an oceanic source of acetaldehyde produced via light-driven photodegradation of CDOM in the
seawater, the model estimation improved significantly, especially over the Red Sea North. With the oceanic source added, modelled
acetaldehyde became slightly overestimated in clean regions, suggesting that the emission rate employed represents an upper limit.
The results indicate that the ocean plays an important role in the atmospheric acetaldehyde budget, under both clean and polluted
conditions. The underestimated acetaldehyde in the model is significant as it will influence the atmospheric budget of e.g. PAN.
As shown in Figure 1, multiple sources and formation pathways need to be considered to better understand the atmospheric budget
of acetaldehyde. Additional laboratory experiments and field measurements are necessary in order to verify all possible
atmospheric formation mechanisms and to improve model simulations.

**Data availability.**

Data will be made available via: https://edmond.mpdl.mpg.de/imeji/

**Author contributions.**

AE and CS performed PTR-ToF-MS measurement and preliminary data processing. NW conducted data analysis and drafted the
article. AP performed EMAC model simulation. EB and LE are responsible for NMHC measurements and data. DD, BH and HF
provided formaldehyde data. Ozone and actinic flux data were contributed by JS and JNC. Methane and carbon monoxide data
were provided by JP. JL designed and realized the campaign. JW supervised the study. All authors contributed to editing the draft
and approved the submitted version.

**Competing interest.**

The authors declare that they have no conflict of interest.

**Acknowledgements**

We acknowledge the collaboration with the King Abdullah University of Science and Technology (KAUST), the Kuwait Institute
for Scientific Research (KISR) and the Cyprus Institute (CyI) to fulfill the campaign. We would like to thank Captain Pavel Kirzner
and the crew for their full support on-board the Kommandor Iona, Hays Ships Ltd,. We are grateful for the support from all
members involved in AQABA campaign, especially Dr. Hartwig Harder for his general organization onboard of the campaign;
and Dr. Marcel Dorf, Claus Koeppel, Thomas Klüpfel and Rolf Hofmann for logistical organization and their help with preparation
and setup. We would like to express our gratitude to Ivan Tadic and Philipp Eger for the use of ship exhaust contamination flag.
Nijing Wang would acknowledges the European Union's Horizon 2020 research and innovation programme under the Marie
Skłodowska-Curie grant agreement No. 674911.






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
