# Peer review of "Measurements of carbonyl compounds around the Arabian Peninsula"

_Atmospheric Chemistry and Physics, 2020_

## Referee Comment (RC1) · Anonymous Referee #1 · 27 Apr 2020

In this work, Nijing Wang and colleagues presented ship-borne measurements of a number of atmospheric carbonyl compounds around the Arabian Peninsula. During the AQABA campaign in 2017, a comprehensive suite of scientific payload provided the simultaneous measurements of an array of trace gases, including the carbonyl compounds measured using a PTR-MS system. The impacts of oceanic emissions and the oil and gas industry in this region are discussed. The manuscript is aligned with the scope of ACP, and the topic is of interest to the community.

A clear focus is given to acetaldehyde in this manuscript. It has been recognized that in situ measurement techniques of acetaldehyde may be subject to substantial inlet

artifacts, especially in remote environments. Such artifacts may be related to tropospherically relevant ozone levels and have been reported for the PTR-MS system on research aircraft platforms (e.g., Northway et al., 2004). Compared to airborne measurements, in situ instruments onboard research vessels usually require much longer sampling lines due to logistic reasons; and the ship may had frequently encountered organic-rich air masses (e.g., polluted air, ship exhaust, and marine air with sea spray aerosols and possibly unsaturated organic compounds). The potential inlet artifacts for the ship-borne trace gas measurements have not been thoroughly discussed in many previous ship-borne studies. In this work, the authors presented fairly detailed results and discussion regarding the potential inlet artifacts, and concluded that the inlet artifacts, although cannot be fully ruled out, are unlikely to be significant in this dataset. I do appreciate the efforts the authors have invested in the potential inlet artifacts. Based on the results presented in this manuscript, I cautiously agree with the authors that the inlet artifacts are probably not a huge concern in this work. More thorough inlet tests would provide more information, which should be considered in future cruise deployments.

In addition, a global chemical transport model (EMAC) is used to examine the sources and sinks of these compounds in this region. The modeling component of this work certainly provides valuable insights. A more thorough model evaluation should also be carried out for some of the key precursors of these carbonyl compounds, such as ethane, propane, butanes, etc.

However, the empirical analysis presented in Section 3.2.1 is largely based on fundamentally flawed assumptions, especially for formaldehyde, acetaldehyde, and acetone. I will elaborate this later. This is one major drawback of this study. For this reason, it is my opinion that major revision is needed before publication in ACP. I would recommend the authors remove this section entirely. Or, perhaps some sort of box modeling (e.g., diurnal steady state model constrained to observed hydrocarbons and other measurements available) may be useful.

The following comments should also been addressed:

Page 1, line 14: "In this study we examine carbonyl compounds (CxHyO), ..." Clearly the formula CxHyO can be an alcohol, an ether, or an epoxide. I do not think a general formula is useful here.

Page 2, line 47: "...important source of free radicals (HOx)" should be hydroxyl radicals here. Also x should be in subscript. Unless the authors have other radicals in mind, in which case it should be more specific.

Page 2, line 48: NOx: please define all abbreviations the first time it appears in the main text. Also x should be in subscript.

Page 2, line 56-57: for the sake of completeness, should cite a few more previous studies here (e.g., Yang et al., 2014; Wang et al., 2019), in which tens of ppt of acetaldehyde were reported in the remote troposphere.

Page 2. Line 59: "... in those regions" this can be confusing. Please revise.

Page 4, line 100: VOC container might be misleading here. I'm guessing this is one of the lab containers loaded aboard, but it could be confused with some sort of container for volatile organic compounds.

Page 5, line 144: please provide a bit more info about this spectral radiometer, such as manufacturer, model, etc.

Page 5, line 145: since NMHCs provide vital constraints on the budget analysis in this work, please provide a bit more information here. For example, is this an online or canister-based GC-FID?

Page 5, line 156: I understand that EMAC is a well established model and many key components are archived elsewhere. Given the importance of oceanic emissions in this work, I am sure the readers would appreciate a bit more details, especially how the oceanic emissions are setup/configured for the compounds discussed in this work,

such as acetaldehyde, acetone, etc.

Page 8, line 210-223: are these carbonyl compounds (i) directly emitted from the oil & gas facilities; (ii) produced from precursors emitted from these facilities; or (iii) associated with other anthropogenic activities in this area?

Page 10, Table 2: I am not entirely sure about the purpose of this table. I understand that the overarching goal here is to put the measurements obtained in this study into the context of previous studies. However, this table itself certainly does not cover the comprehensive characteristics of any of the category. For example, some of the studies cited in the marine category contains clear influence from anthropogenic emissions/biomass burning; none of the cited studies in this category reflects the pristine marine environments, such as tropics, Southern Ocean, etc. For this reason, this table may create wrong impression to the audience. I understand that this is not a review paper, but a more thorough review of the literature is always appreciated. Therefore, I would suggest that the authors compile a slightly more exhaustive list, and be cautious when quoting/discussing the numbers in the literature. I myself find some of the categories are less relevant for this work (e.g., urban, rural, forest), and the marine category can certainly use more details (e.g., coastal vs open ocean, northern vs southern hemisphere, high latitude oceans, natural vs influenced by anthropogenic/biomass burning emissions). I would also recommend the authors add relevant info (e.g., lat/lon) to this table.

Page 9, line 236-237: why do the authors compare the ship-borne measurements obtained in this work to the measurements from a rural site in Cyprus? The numbers may be similar for vastly different reasons. Or do the authors imply a connection with Cyprus? If yes, please clarify & elaborate.

Page 9, line 247-248: note that acetone may deposit into the ocean, especially in the continental outflow from polluted regions (e.g., Marandino et al., 2005; Schlundt et al., 2017; Yang et al., 2014).

Page 9, line 253: Again, the authors compare the results in Red Sea to that from the other side of the planet (Thompson Farm, a rural site in the US). What is the point of the seemingly random comparison?

Page 11, line 269: are these numbers the sum of all measured unsaturated and aromatic carbonyls?

Page 11, line 293-295: these very general descriptions belong to the introduction section.

Page 11, line 298-299: The first two assumptions do not hold, for the following reasons: (1) Photolysis can be a significant sink for certain carbonyl compounds, such as acetone (Fischer et al., 2012); for formaldehyde it's even the dominant sink (Anderson et al., 2017). (2) Primary emissions contribute substantially to many carbonyl compounds in the atmosphere, such as acetone (e.g., Fischer et al., 2012), acetaldehyde (e.g., Millet et al., 2010). Biomass burning contribution to many of these carbonyl compounds is also substantial, and the authors even discussed the impacts of biomass burning in Section 3.2.2. Therefore, primary emissions certainly cannot and should not be ignored. (3) Dilution (mixing with background air) can lead to rapid decrease of tracer concentrations in freshly emitted plumes. Uptake by the terrestrial vegetation and the ocean may also affect the carbonyl budget on the regional and global scale. The impact of mixing may be cancelled out by scaling the mixing ratios of hydrocarbons and carbonyls to a long-lived compound, such as CO or ethylene, but I cannot think of a simple way to overcome other impacts as I listed above. In addition to the flawed assumptions, the carbonyl yields are not constant and depend on other variables (e.g., NOx levels, oxidation pathways). Table S4 did not clarify how the yields were derived.

Page 12, line 314-319: This exercise is only valid when the ratio in the source region is well understood, which is clearly not the case here. Figure S3 cannot rule out the possibility that there may be air masses with different initial toluene/benzene ratios mixed together. It certainly looks like there are multiple clusters in Figure S3. This

comes back to my previous question regarding the oil and gas industry in this region: do these measurements reflect (i) direct emissions from the oil & gas facilities; or (ii) co-located (anthropogenic) emissions in this area? This type of photochemical clock does not work without a thorough understanding of the source characteristics.

To sum up, the OH exposure calculation may be useful, provided that the source contributions are well understood. The empirical analysis based on Equation 1 and 2, however, does not really provide scientifically valuable information, definitely not for acetone, acetaldehyde, and formaldehyde.

Page 14, Figure 4: Units for j_NO2 and OH exposure are missing. Same for Figure 5.

Page 14, line 366: "As mentioned before, photochemical oxidation contributed a large fraction to acetone and the larger aliphatic carbonyls over the Arabian Gulf and Suez areas" I think this statement needs stronger support. The terrestrial biosphere may emit a large amount of acetone. Moreover, the acetone yield from terpene oxidation is quite high, and the acetone production from terpenes may not be a huge source globally, but could well be in certain regions. Similarly, some of these compounds may be from anthropogenic sources or biomass burning.

Page 14, line 374-376: The authors concluded that the carbonyls are co-produced via photochemical oxidation because they show strong correlation with ozone. I am not totally convinced. It is plausible that these carbonyls may be co-emitted with ozone precursors.

Page 14, line 358: I noticed that the spatial distribution oil fields and refineries is from the year of 2007, while this cruise campaign was conducted in 2017. Is it possible to obtain more recent information on this?

Page 15, line 405: "The biomass burning emissions were probably transported by on the prevailing northerly wind above Northeast Egypt..." is this supported by backward trajectory analysis or meteorological conditions?

Page 16, Figure 6: this is interesting. how are these not-assigned ions calibrated/quantified?

Page 17, line 449-452: any GC-FID measurements of isoprene during this period?

Page 17, line 460-462: "The model considers direct emissions (such as anthropogenic, biogenic, biomass burning etc.), atmospheric transport and mixing, photochemical production of carbonyls (by OH, O3 and NO3), and physical and chemical removal processes." This belongs to the method section where the model is introduced.

Page 17, line 465-466: "acetone was overestimated by a factor within 1.5 over the Arabian Sea, Gulf of Aden and Gulf of Oman, and underestimated by a factor within 2.5 over the other regions." Note that previous studies reported downward oceanic fluxes of acetone (ocean being a net sink) in the continental outflow from polluted regions (e.g., Marandino et al., 2005; Schlundt et al., 2017; Yang et al., 2014). How does the model treat the air-sea exchange of acetone? This key information is missing.

Page 19, line 504-505: "This indicates that the source of acetaldehyde was probably not from direct biogenic production..." this sentence is confusing. I think "direct oceanic acetaldehyde emissions are probably insufficient to explain..." might be more accurate?

Page 19, line 513-514: "To our knowledge, there is no clear experimental evidence showing the ocean to be a sink for acetaldehyde." This is probably true, but the language is vague (I think "direct" is a better choice than "clear" here), and several important studies are missing here. Schlundt et al. (2017) reported net downward fluxes of acetaldehyde in the polluted marine boundary layer (ocean is a net sink), which is inferred from measurements in the atmosphere and the surface ocean (so not "direct" evidence). Yang et al. (2014) reported oceanic fluxes of acetaldehyde using eddy covariance method (this is direct flux measurements). Indeed the fluxes were mostly upward for acetaldehyde (ocean is a net source), there appears to be a few points indicative of net downward fluxes, although are probably close to the limit detection of

that system.

Page 21, line 549-554: can the authors please provide new plots like Figure 7 but for other major acetaldehyde/acetone precursors, such as ethane, propane, butanes, and perhaps ethanol as well? This would certainly make the case stronger, and may provide key insight into the acetaldehyde budget in this region. Even "remote" regions in this work are not really that far from the source regions. Therefore it remains unclear to what degree the underestimation/overestimation of carbonyls is due to their precursors. Emission inventories often have a hard time capturing the anthropogenic emissions, especially from the oil and gas industry.

Page 21, line 570: what biomass burning emission inventory is used in this work?

References

Anderson, D. C., Nicely, J. M., Wolfe, G. M., Hanisco, T. F., Salawitch, R. J., Canty, T. P., Dickerson, R. R., Apel, E. C., Baidar, S., Bannan, T. J., Blake, N. J., Chen, D., Dix, B., Fernandez, R. P., Hall, S. R., Hornbrook, R. S., Gregory Huey, L., Josse, B., Jöckel, P., Kinnison, D. E., Koenig, T. K., Le Breton, M., Marécal, V., Morgenstern, O., Oman, L. D., Pan, L. L., Percival, C., Plummer, D., Revell, L. E., Rozanov, E., Saiz-Lopez, A., Stenke, A., Sudo, K., Tilmes, S., Ullmann, K., Volkamer, R., Weinheimer, A. J. and Zeng, G.: Formaldehyde in the Tropical Western Pacific: Chemical Sources and Sinks, Convective Transport, and Representation in CAM-Chem and the CCMI Models, Journal of Geophysical Research: Atmospheres, 122(20), 11,201–211,226, doi:10.1002/2016JD026121, 2017.

Brewer, J. F., Bishop, M., Kelp, M., Keller, C. A., Ravishankara, A. R. and Fischer, E. V.: A sensitivity analysis of key natural factors in the modeled global acetone budget, Journal of Geophysical Research: Atmospheres, 122(3), 2043–2058, doi:10.1002/2016JD025935, 2017.

Fischer, E. V., Jacob, D. J., Millet, D. B., Yantosca, R. M. and Mao, J.: The role of

the ocean in the global atmospheric budget of acetone, Geophysical Research Letters, 39(1), 2012.

Marandino, C. A., Bruyn, W. J. D., Miller, S. D., Prather, M. J. and Saltzman, E. S.: Oceanic uptake and the global atmospheric acetone budget, Geophysical Research Letters, 32(15), doi:10.1029/2005GL023285, 2005.

Millet, D. B., Guenther, A., Siegel, D. A., Nelson, N. B., Singh, H. B., de Gouw, J. A., Warneke, C., Williams, J., Eerdekens, G. and Sinha, V.: Global atmospheric budget of acetaldehyde: 3-D model analysis and constraints from in-situ and satellite observations, Atmos. Chem. Phys, 10(7), 3405–3425, 2010. Northway, M. J., de Gouw, J. A., Fahey, D. W., Gao, R. S., Warneke, C., Roberts, J. M. and Flocke, F.: Evaluation of the role of heterogeneous oxidation of alkenes in the detection of atmospheric acetaldehyde, Atmospheric Environment, 38(35), 6017–6028, doi:https://doi.org/10.1016/j.atmosenv.2004.06.039, 2004.

Schlundt, C., Tegtmeier, S., Lennartz, S. T., Bracher, A., Cheah, W., Krüger, K., Quack, B. and Marandino, C. A.: Oxygenated volatile organic carbon in the western Pacific convective center: ocean cycling, air–sea gas exchange and atmospheric transport, Atmos. Chem. Phys., 17(17), 10837–10854, doi:10.5194/acp-17-10837-2017, 2017.

Wang, S., Hornbrook, R. S., Hills, A., Emmons, L. K., Tilmes, S., Lamarque, J.-F., Jimenez, J. L., Campuzano‐Jost, P., Nault, B. A., Crounse, J. D., Wennberg, P. O., Ryerson, T. B., Thompson, C. R., Peischl, J., Moore, F., Nance, D., Hall, B., Elkins, J., Tanner, D., Huey, L. G., Hall, S. R., Ullmann, K., Orlando, J. J., Tyndall, G. S., Flocke, F. M., Ray, E., Hanisco, T. F., Wolfe, G. M., Clair, J. S., Commane, R., Daube, B., Barletta, B., Blake, D. R., Weinzierl, B., Dollner, M., Conley, A., Vitt, F., Wofsy, S. C., Riemer, D. D. and Apel, E. C.: Atmospheric Acetaldehyde: Importance of Air-Sea Exchange and a Missing Source in the Remote Troposphere, Geophysical Research Letters, doi:10.1029/2019GL082034, 2019.

Yang, M., Beale, R., Liss, P., Johnson, M., Blomquist, B. and Nightingale, P.: Air–sea

fluxes of oxygenated volatile organic compounds across the Atlantic Ocean, Atmospheric Chemistry and Physics, 14(14), 7499–7517, doi:10.5194/acp-14-7499-2014, 2014.

---

## Referee Comment (RC2) · Anonymous Referee #2 · 27 May 2020

Comments to "Measurements of carbonyl compounds around the Arabian Peninsula indicate large missing sources of acetaldehyde" by Wang et al.

**Overview:**

The manuscript presented ship-borne measurements of carbonyl compounds (carbonyls) around the Arabian Peninsula during the AQABA campaign (end of June to end of August 2017) and interpreted the measurement with a chemical transport model (EMAC). Analyses were focused on 3 classified carbonyl groups—aliphatic ($C_nH_{2n}O$), aromatic ($C_nH_{2n-8}O$), and unsaturated ($C_nH_{2n-2}O$)—in terms of their regional distribution, source characterization and partitioning, model-measurement comparison in 8 geographical regions, and finally an in-depth discussion on the missing sources of acetaldehyde (C2 aliphatic carbonyl).

The manuscript is well organized with clear methods, thorough analyses, and coherent arguments. The topic is of great importance given that carbonyls can provide key clues on reactive carbon source and chemistry as an oxygenated and stable group of VOCs, and acetaldehyde has drawn increased attention recently with growing evidence of its large missing sources. This manuscript therefore provided valuable information in advancing the knowledge of atmospheric reactive carbon in the Middle East. I support the publication of the manuscript once the following issues being addressed.

**Scientific comments:**

1)  The title is interesting but over-weights the topic towards acetaldehyde and meanwhile lacks a reflection of the large portion of work done on the overview analysis of different carbonyls in this region as presented in the manuscript.
2)  L22. "A smaller but still significant discrepancy …", are you trying to say that the model-measurement discrepancy of other carbonyls is smaller comparing to that of acetaldehyde? This sentence is not clear and needs to be reorganized.
3)  L130. How did you deal with below LOD data?
4)  L134-140. There are other measurement techniques that can differentiate ketone and aldehyde at a shared mass, better to give an average or a range of in-situ reported ketone:aldehyde ratio (e.g., acetone:propanal ratio) to prove the dominance of ketone.
5)  Figure2. The description of air mass characteristics is not clear. For example, what's the difference between "source nearby" vs "source nearby, aged", "remote, aged" vs "remote, clean"?
6)  L246-247. "Interestingly, the mean acetaldehyde mixing ratio (0.62 ± 0.59 ppb) over Suez was twice the level found over the Mediterranean Sea, whilst the acetone level was only slightly higher", why is this interesting? What are you trying to imply?
7)  L247-248. "Besides the local-scale emission and photochemical production contribution to the acetone over Suez, the longer lived acetone could be also transported from the Mediterranean Sea (where acetone was high)". Acetone is quite similar in Mediterranean and Suez. "where acetone was high" seems to indicate that acetone source in Suez is dominated by transport rather than local production. We do not know the contribution of transport vs. local source to acetone in Suez region, so need to provide evidence or reorganize the sentence.

8) L249-250. "Although the mean mixing ratios of aliphatic carbonyls over Suez were lower than those over the Arabian Gulf, larger variations were observed." I'm not seeing the higher variation over Suez than Arabian Gulf.

9) L254-258. The arguments here are based on back trajectory. How about the proposed deep sea local source of ethane and propane in Bourtsoukidis et al. 2020? This was found not sufficient to explain the model bias of acetaldehyde later in the paper, but it worth being brought up here as one potential explanation of high carbonyls in the Red Sea North and also briefly mention what you found later in the model …

10) Table 2. Are you summarizing only surface or tower measurement? Need to clarify.
Too few examples of Oil & Gas region and Forest, add more.

11) L291, the title of Section 3.2 "Chemistry of aliphatic carbonyls", does not reflect the source characterization discussed in the two case studies in Section 3.2.2.

12) L303, Eq. (2). Varying formats of this equation have been widely applied but I have one question regarding its application in secondary OVOCs from aged air. Eq. (2) assumes that both precursor hydrocarbon and aliphatic carbonyls experienced the same degree of OH exposure (or Δt). This assumption is reasonable for fresh air mass, but might not be valid for aged air, in which carbonyls kept being produced along the way (depending on its precursor's lifetime) and thus experienced different degrees of OH exposure. So, for long-transported air masses, this could lead to an overestimated OH exposure, an underestimated carbonyl secondary production, and thus an underestimated fractional contribution from oxidation to the measured carbonyl mixing ratio.

Would this uncertainty affect your conclusion (P12, L334-336; P22, L615-618) that hydrocarbon oxidation plays a more important role in polluted regions than other less-polluted regions, given the potential low bias in the estimated contribution for less-polluted/more-remote regions?

13) L308-312. Table S4. How is the yield being derived from MCM?

14) L318. Did you use 0-5AM data as in Yuan et al. 2012? Clarify that.

L319. How do your calculated emission ratios compare to literature reported values from similar sources?

15) L329, "In general, the direct oxidation fraction varied from area to area for C1 to C3 carbonyls (formaldehyde, acetaldehyde and acetone)." Give the range of fractions for each species.

16) L414, Section 3.3 title "air chemistry of unsaturated carbonyls" does not reflect the discussion in this section, which is more focused on potential precursor/source of large unsaturated carbonyls.

17) L470-471. "A strong natural non methane hydrocarbon source from deep water in the Northern Red Sea was implemented in the model (Bourtsoukidis et al., 2020)." This needs to be added in the method section where you describe the model configuration. There you have stated that the configuration is following Bourtsoukidis et al. 2020, but also worth pointing out that this newly proposed source is applied.

18) L520. "the air-sea submodel (Pozzer et al., 2006) was augmented to include …" what is the air-sea submodel? Is that a model coupled into the EMAC model? Or it's a separate model you used to get the ocean acetaldehyde concentration field? Need to clarify. And if it's coupled to EMAC, need to add in the EMAC model configuration section.

19) P21, Section 3.5.3, "Anthropogenic primary sources". Comparing to Section 3.5.1 and 3.5.2, this section is lack of analysis-based discussion.

Also, if the revised version kept the same length or longer, better to separate into two paragraphs, with the first one discussing the underestimated anthropogenic primary emissions of acetaldehyde precursors (L550-558) and the second talking about underestimated anthropogenic primary emissions of acetaldehyde itself (L558-end).

L551. Does the carbon number of unmeasured hydrocarbons start from C12? Figure 3 and 6 only considered HCs up to C8.

L552-555. "Bourtsoukidis et al. (2020) compared measured hydrocarbons (ethane, propane, ethene etc.) with the results from model simulations (the same model used in this study) and periodically found significant model underestimation in both regions. This indicates that not all sources were present in the model's emission inventory". This seems to argue that the low bias of ethane, propane, ethene, etc. in the model inventory is one reason for the model underestimation of acetaldehyde, which contradicts the argument (P13, L340-349) based on Figure 3. The argument of non-measured and non-modeled HCs before this sentence is fine, but this sentence makes it unclear.

L555-550. 1) need to say that the high ozone in case studies was observed during "nighttime". 2) Does the modeled occasional underestimate of ethene persist in nighttime too?

L565-567. "Therefore, the active petroleum industry located in the Arabian Gulf and intensive marine transportation in Suez are likely primary sources of acetaldehyde and other carbonyls which were not well constrained in the model." This is not convincing.

20) L587, "Another acetaldehyde formation pathway …" suggest starting a new paragraph to discuss pyruvic acid.

L590-592. "Zhou and Mopper (1997) pointed out that the net exchange direction for pyruvic acid is expected to be from the air to the sea due to its high partition coefficient (high solubility). Therefore, only low levels of pyruvic acid would be expected in the remote marine boundary layer". High air-to-sea partition tendency does not guarantee low level of pyruvic acid in the air-side, if gas-liquid partitioning is the only limiting process, its concentration in the air also depends on its degree of saturation in the surface seawater. A review of surface seawater concentration of pyruvic acid and an air-sea equilibrium calculation might be helpful.

L597-598. "As the air-sea exchange of pyruvic acid is limited, low levels of pyruvic acid were expected" Not clear what this means. How did you know that the air-sea exchange of pyruvic acid is limited? And why would that result in low levels of pyruvic acid?

L598-599. "Even if we fully assign the m/z 89.024 to pyruvic acid, the contribution to acetaldehyde via photolysis of pyruvic acid is negligible compared other sources" Clarify why it is negligible, using 58 ppt and a 100% yield? Is this contribution detectable by intrument?

**Minor/technical comments:**

L17: typo "3,8"

L89: section numbering error, should start with 2.1

L131 and Table S1: 3σ in text but 1σ in Table S1

L135: add "of" between "number" and "three"

L156: "interaction with ocean/land" includes "dry deposition"

Table 1. L193, L207. "Aliphatic CCs, Aromatic CCs, Unsaturated CCS", "CC" hasn't been defined.

Table 1. L193, L207. Extend the name of "S" to "Suez"

L348. "section" to "sections"

Figures 4, 5, and 7. Change x-axis label from "Dateandtime" to "Dateandtime (UTC)", or clarity in the figure caption.

Figures 4 and 5. missing units for several y-axis, "J_NO2", "[OH]t", "Wind_D"

Figure 7a left panel. "S" to "Suez"

Figure S6. Only 6 regions shown in the figure.

---

## Author Comment (AC1) · 17 Jul 2020

Dear Referee,

Thank you very much for you thoughtful and constructive comments for our work. We have addressed them all in the revised version of the manuscript as described below:

**In this work, Nijing Wang and colleagues presented ship-borne measurements of a number of atmospheric carbonyl compounds around the Arabian Peninsula. During the AQABA campaign in 2017, a comprehensive suite of scientific payload provided the simultaneous measurements of an array of trace gases, including the carbonyl compounds measured using a PTR-MS system. The impacts of oceanic emissions and the oil and gas industry in this region are discussed. The manuscript is aligned with the scope of ACP, and the topic is of interest to the community.**

**A clear focus is given to acetaldehyde in this manuscript. It has been recognized that in situ measurement techniques of acetaldehyde may be subject to substantial inlet artifacts, especially in remote environments. Such artifacts may be related to tropospherically relevant ozone levels and have been reported for the PTR-MS system on research aircraft platforms (e.g., Northway et al., 2004). Compared to airborne measurements, in situ instruments onboard research vessels usually require much longer sampling lines due to logistic reasons; and the ship may had frequently encountered organic-rich air masses (e.g., polluted air, ship exhaust, and marine air with sea spray aerosols and possibly unsaturated organic compounds). The potential inlet artifacts for the ship-borne trace gas measurements have not been thoroughly discussed in many previous ship-borne studies. In this work, the authors presented fairly detailed results and discussion regarding the potential inlet artifacts, and concluded that the inlet artifacts, although cannot be fully ruled out, are unlikely to be significant in this dataset. I do appreciate the efforts the authors have invested in the potential inlet artifacts. Based on the results presented in this manuscript, I cautiously agree with the authors that the inlet artifacts are probably not a huge concern in this work. More thorough inlet tests would provide more information, which should be considered in future cruise deployments.**

We thanks the reviewer for noting and agreeing with our assessment of the potential inlet artifact. Through this work we have become more aware of this issue and will characterize the inlet accordingly in future measurement campaigns.

**In addition, a global chemical transport model (EMAC) is used to examine the sources and sinks of these compounds in this region. The modeling component of this work certainly provides valuable insights. A more thorough model evaluation should also be carried out for some of the key precursors of these carbonyl compounds, such as ethane, propane, butanes, etc.**

A thorough model evaluation of the major alkanes measured during AQABA has been published in *Nature Communications* earlier this year (Bourtsoukidis et al., 2020)., The aforementioned study also included multiple sensitivity analyses by the model to investigate alkane sources. We therefore refer to the published work in this manuscript to avoid duplication. . We have further addressed the model comparison results in terms of carbonyl precursors in section 3.4.3.

**However, the empirical analysis presented in Section 3.2.1 is largely based on fundamentally flawed assumptions, especially for formaldehyde, acetaldehyde, and acetone. I will elaborate this later. This is one major drawback of this study. For this reason, it is my opinion that major revision is needed**

**before publication in ACP. I would recommend the authors remove this section entirely. Or, perhaps some sort of box modeling (e.g., diurnal steady state model constrained to observed hydrocarbons and other measurements available) may be useful.**

On reflection, we agree with the reviewer that this section did not contribute significantly to the analysis and we decided to remove this section entirely.

Detailed response:

**Page 1, line 14: "In this study we examine carbonyl compounds (CxHyO), …" Clearly the formula CxHyO can be an alcohol, an ether, or an epoxide. I do not think a general formula is useful here.**

We agree. CxHyO is removed in the text.

**Page 2, line 47: "…important source of free radicals (HOx)" should be hydroxyl radicals here. Also x should be in subscript. Unless the authors have other radicals in mind, in which case it should be more specific.**

We revised to "hydroxyl and hydroperoxyl radicals free radicals ($HO_x$)" in the text.

**Page 2, line 48: NOx: please define all abbreviations the first time it appears in the main text. Also x should be in subscript.**

We revised to "$NO_x$ (NO and $NO_2$)" in the text.

**Page 2, line 56-57: for the sake of completeness, should cite a few more previous studies here (e.g., Yang et al., 2014; Wang et al., 2019), in which tens of ppt of acetaldehyde were reported in the remote troposphere.**

More references were added in the text:

Remote area:

Wisthaler, A.: Organic trace gas measurements by PTR-MS during INDOEX 1999, Journal of Geophysical Research, 107, 10.1029/2001jd000576, 2002.

Yang, M., Beale, R., Liss, P., Johnson, M., Blomquist, B., and Nightingale, P.: Air–sea fluxes of oxygenated volatile organic compounds across the Atlantic Ocean, Atmospheric Chemistry and Physics, 14, 7499-7517, 10.5194/acp-14-7499-2014, 2014.

Wang, S., Hornbrook, R. S., Hills, A., Emmons, L. K., Tilmes, S., Lamarque, J. F., Jimenez, J. L., Campuzano‐Jost, P., Nault, B. A., Crounse, J. D., Wennberg, P. O., Kim, M., Allen, H., Ryerson, T. B., Thompson, C. R., Peischl, J., Moore, F., Nance, D., Hall, B., Elkins, J., Tanner, D., Huey, L. G., Hall, S. R., Ullmann, K., Orlando, J. J., Tyndall, G. S., Flocke, F. M., Ray, E., Hanisco, T. F., Wolfe, G. M., St. Clair, J., Commane, R., Daube, B., Barletta, B., Blake, D. R., Weinzierl, B., Dollner, M., Conley, A., Vitt, F., Wofsy, S. C., Riemer, D. D., and Apel, E. C.: Atmospheric Acetaldehyde: Importance of Air‐Sea

Exchange and a Missing Source in the Remote Troposphere, Geophysical Research Letters, 10.1029/2019gl082034, 2019.

Polluted area:

Koss, A. R., de Gouw, J., Warneke, C., Gilman, J. B., Lerner, B. M., Graus, M., Yuan, B., Edwards, P., Brown, S. S., Wild, R., Roberts, J. M., Bates, T. S., and Quinn, P. K.: Photochemical aging of volatile organic compounds associated with oil and natural gas extraction in the Uintah Basin, UT, during a wintertime ozone formation event, Atmospheric Chemistry and Physics, 15, 5727-5741, 10.5194/acp-15-5727-2015, 2015.

**Page 2. Line 59: "… in those regions" this can be confusing. Please revise.**

We revised to "Generally, secondary photochemical formation from various precursors is the main source for those carbonyl compounds." in the text to avoid the confusion.

**Page 4, line 100: VOC container might be misleading here. I'm guessing this is one of the lab containers loaded aboard, but it could be confused with some sort of container for volatile organic compounds.**

We revised "VOC container" to "on-board lab container" in the text.

**Page 5, line 144: please provide a bit more info about this spectral radiometer, such as manufacturer, model, etc.**

The manufacturer and related reference were added in the text as "The actinic flux was measured by a spectral radiometer (Metcon GmbH; Meusel et al., 2016)."

**Page 5, line 145: since NMHCs provide vital constraints on the budget analysis in this work, please provide a bit more information here. For example, is this an online or canister-based GC-FID?**

More information about NMHCs measurement was added in the text as "Non methane hydrocarbons (NMHC) mixing ratios were measured by a gas chromatograph with flame ionization detector (GC−FID) online with the time resolution of 50 minutes. It measured hydrocarbons (C2 - C8) and aromatics (C6 - C8) with the average LOD < 10 ppt for most of compounds. For a detailed instrumental description see Bourtsoukidis et al. (2019)."

**Page 5, line 156: I understand that EMAC is a well established model and many key components are archived elsewhere. Given the importance of oceanic emissions in this work, I am sure the readers would appreciate a bit more details, especially how the oceanic emissions are setup/configured for the compounds discussed in this work, such as acetaldehyde, acetone, etc.**

More information on EMAC and in particular the air-sea transfer was added in the text as follows "… a comprehensive chemistry mechanism MOM (Mainz Organic Mechanism) was deployed (Sander et al., 2019). The model considers direct emissions (such as anthropogenic, biogenic, biomass burning etc.), atmospheric transport and mixing, photochemical production of carbonyls (by OH, $O_3$ and $NO_3$), as well as physical and chemical removal processes. The exchange of organic compounds between ocean and atmosphere was considered in EMAC via the AIRSEA submodel, described in detail in Pozzer et al. (2006). The transfer velocity is calculated online and the concentration in the water is prescribed by the user. For acetone, a constant water concentration of 15 nmol/L is used, following the suggestion of Fischer et al. (2012), while direct oceanic emission of acetaldehyde was initially not included in the model. The model configuration in the study is the same as the model applied in Bourtsoukidis et al. (2020) in the resolution of T106L31 (i.e. ~ $1.1^\circ \times 1.1^\circ$ horizontal resolution and , 31 vertical hybrid pressure levels up to 10 hPa) and the time resolution of 10 minutes. The measurement data of PTR-ToF-MS were averaged to 10-minute resolution to match the model data resolution for further comparison."

**Page 8, line 210-223: are these carbonyl compounds (i) directly emitted from the oil & gas facilities; (ii) produced from precursors emitted from these facilities; or (iii) associated with other anthropogenic activities in this area?**

We discussed these sources (first or secondary formation of carbonyls from anthropogenic activities) in section 3.1.2 (case studies of polluted regions) and also section 3.4.3 (anthropogenic primary sources for underestimated acetaldehyde). In order to make this clear, we now added a sentence in the end of this paragraph as follow: The sources of the major carbonyls in the Arabian Gulf will be discussed in details in section 3.1.2 and 3.4.3.

**Page 10, Table 2: I am not entirely sure about the purpose of this table. I understand that the overarching goal here is to put the measurements obtained in this study into the context of previous studies. However, this table itself certainly does not cover the comprehensive characteristics of any of the category. For example, some of the studies cited in the marine category contains clear influence from anthropogenic emissions/ biomass burning; none of the cited studies in this category reflects the pristine marine environments, such as tropics, Southern Ocean, etc. For this reason, this table may create wrong impression to the audience. I understand that this is not a review paper, but a more thorough review of the literature is always appreciated. Therefore, I would suggest that the authors compile a slightly more exhaustive list, and be cautious when quoting/discussing the numbers in the literature. I myself find some of the categories are less relevant for this work (e.g., urban, rural, forest), and the marine category can certainly use more details (e.g., coastal vs open ocean, northern vs southern hemisphere, high latitude oceans, natural vs influenced by anthropogenic/biomass burning emissions). I would also recommend the authors add relevant info (e.g., lat/lon) to this table.**

We now updated the table by including 5 more studies done in open sea and coastal areas. We replaced the study of oil and gas influenced regions by other four studies. Studies done in urban, rural and forest regions were removed from the table. We included the coordinates (latitude and longitude) and measurement sampling height for those studies as requested. Accordingly, we updated the text related to the comparison of previous studies in the revised manuscript. The revised table is attached in the end of this response.

**Page 9, line 236-237: why do the authors compare the ship-borne measurements obtained in this work to the measurements from a rural site in Cyprus? The numbers may be similar for vastly different reasons. Or do the authors imply a connection with Cyprus? If yes, please clarify & elaborate.**

As suggested by the reviewer, the table has been updated and the measurements results from this Cyprus study as well as the related text were removed and rewritten in the revised manuscript as: "… The Mediterranean Sea had somewhat higher levels of aliphatic carbonyls than the clean regions (the Arabian Sea and the Gulf of Aden) but with acetone (above 2ppb) dominating the distribution. Much higher acetone level than acetaldehyde level was also observed for some costal site measurement which was impacted by continental air (White et al., 2008; Schlundt et al., 2017, see Table 2). …"

**Page 9, line 247-248: note that acetone may deposit into the ocean, especially in the continental outflow from polluted regions (e.g., Marandino et al., 2005; Schlundt et al., 2017; Yang et al., 2014).**

We are aware that under conditions of polluted continental outflow over the ocean uptake of acetone from the air to the sea will occur. Also dilution and mixing with free tropospheric air during transport can modulate acetone mixing ratios. This process is accounted for in the EMAC model used in the subsequent analysis. We now added a sentence in the revised manuscript to make it clear as follow: "… could be also transported from the Mediterranean Sea (where acetone was high). Four-day back trajectories indicate the air reaching Suez region was mostly originated from Europe continent passing over the Mediterranean Sea (Bourtsoukidis et al., 2019). Meanwhile, ocean uptake of acetone from the air due to polluted continental outflow (Marandino et al., 2005) as well as dilution and mixing with free tropospheric air during transport can modulate acetone mixing ratios."

**Page 9, line 253: Again, the authors compare the results in Red Sea to that from the other side of the planet (Thompson Farm, a rural site in the US). What is the point of the seemingly random comparison?**

With these comparisons we were trying to set the AQABA measurements in a broader context. Since this appears to have been a distraction, we have followed the reviewer's suggestion, and the table has been updated, so that the measurements from the Thompson Farm study as well as the related text were removed in the revised manuscript as follows:"…The mixing ratios of acetaldehyde and acetone over the northern part of the Red Sea were similar to those levels measured in western Pacific coastal regions (South China Sea, Table 2)…."

**Page 11, line 269: are these numbers the sum of all measured unsaturated and aromatic carbonyls?**

The numbers represent the range of the mean mixing ratios of unsaturated carbonyls in each region. To make this clear we added the sentence "…followed by Suez (11 – 68 ppt). The numbers represent the range of the mean mixing ratios of unsaturated carbonyls in each region"

**Page 11, line 293-295: these very general descriptions belong to the introduction section.**

These lines were removed in the revised manuscript.

**Page 11, line 298-299: The first two assumptions do not hold, for the following reasons: (1) Photolysis can be a significant sink for certain carbonyl compounds, such as acetone (Fischer et al., 2012); for formaldehyde it's even the dominant sink (Anderson et al., 2017). (2) Primary emissions contribute substantially to many carbonyl compounds in the atmosphere, such as acetone (e.g., Fischer et al., 2012), acetaldehyde (e.g., Millet et al., 2010). Biomass burning contribution to many of these carbonyl compounds is also substantial, and the authors even discussed the impacts of biomass burning in Section 3.2.2. Therefore, primary emissions certainly cannot and should not be ignored. (3) Dilution (mixing with background air) can lead to rapid decrease of tracer concentrations in freshly emitted plumes. Uptake by the terrestrial vegetation and the ocean may also affect the carbonyl budget on the regional and global scale. The impact of mixing may be cancelled out by scaling the mixing ratios of hydrocarbons and carbonyls to a long-lived compound, such as CO or ethylene, but I cannot think of a simple way to overcome other impacts as I listed above. In addition to the flawed assumptions, the carbonyl yields are not constant and depend on other variables (e.g., NOx levels, oxidation pathways). Table S4 did not clarify how the yields were derived.**

Thanks for pointing out the inappropriate application of the empirical method we applied in our study. The intention was to show what the photochemistry alone (without these influences) would generate. We agree that because of the complexity of multiple alternate sources this section does not have much scientific contribution to the whole manuscript. As we compared the measurement with a comprehensive global model in the manuscript (which includes such sources), we removed the part regarding the empirical calculations and corrected relevant text in the revised manuscript. For the yield in Table S4, we assumed the alkylperoxy radicals ($RO_2$) would mainly react with NO (high NO condition) and then summed up the yield of all the reaction pathways of hydrocarbons that could produce carbonyls. Now as we decided to remove the empirical calculation section, the Table S4 was also removed from the supporting information.

**Page 12, line 314-319: This exercise is only valid when the ratio in the source region is well understood, which is clearly not the case here. Figure S3 cannot rule out the possibility that there may be air masses with different initial toluene/benzene ratios mixed together. It certainly looks like there are multiple clusters in Figure S3. This comes back to my previous question regarding the oil and gas industry in this region: do these measurements reflect (i) direct emissions from the oil & gas facilities; or (ii) co-located (anthropogenic) emissions in this area? This type of photochemical clock does not work without a thorough understanding of the source characteristics. To sum up, the OH exposure calculation may be useful, provided that the source contributions are well understood. The empirical analysis based on Equation 1 and 2, however, does not really provide scientifically valuable information, definitely not for acetone, acetaldehyde, and formaldehyde.**

We agree that it is not appropriate to use the hydrocarbon ratios to calculate the OH exposure for the remote areas. However, we kept the estimations of OH exposure using hydrocarbon ratios for the polluted regions (Arabian Gulf and Suez) where the major primary emissions were identified based on NMHCs data reported by Bourtsoukidis et al., (2019) and Bourtsoukidis et al. (2020). We now mention this specifically in the section 3.1.2 in the revised manuscript and also compared the toluene to benzene emission ratios to previous studies having similar sources. The revised text is as follows:

…For further insight, we focused on a time series of selected trace-gases and their inter-correlations to better identify the sources of the major aliphatic carbonyls. Meanwhile, we calculated the OH exposure ($[OH]\Delta t$) based on hydrocarbon ratios (Roberts et al., 1984;de Gouw et al., 2005;Yuan et al., 2012) for the polluted regions Arabian Gulf and Suez where primary emissions have been identified (Bourtsoukidis et al., 2019; Bourtsoukidis et al. 2020), to better understand the photochemical aging of the major carbonyls using the following equation:

$$[OH]\Delta t = \frac{1}{k_X - k_Y} \cdot (In \left.\frac{[X]}{[Y]}\right|_{t=0} - In \frac{[X]}{[Y]}) ,$$  Eq. (1)

where X and Y refer to two hydrocarbon compounds with different rates of reaction with the OH radical (k). For this study, we chose toluene ($k_{OH+toluene}$: 5.63E-12 cm$^3$ molecule$^{-1}$s$^{-1}$ ) and benzene ($k_{OH+benzene}$: 1.22E-12 cm$^3$ molecule$^{-1}$s$^{-1}$) (Atkinson and Arey, 2003), because both compounds were measured by PTR-ToF-MS at high frequency and these values showed a good agreement with values measured by GC-FID (Figure S2). The approach detailed by Yuan et al. (2012) was applied to determine the initial emission ratio $\left.\frac{[X]}{[Y]}\right|_{t=0}$ in those two regions by only including nighttime data of benzene and toluene. We obtained initial emission ratios (toluene to benzene ratio) of 1.38 for the Arabian Gulf and 2.12 for the Suez region. Koss et al. (2017) summarized the toluene to benzene ratios observed in various locations and showed that urban and vehicle sources tend to have higher toluene to benzene ratio (mean ~ 2.5) than the ratios of oil & gas sources (mean ~ 1.2). Therefore, the toluene to benzene ratios obtained for those two regions agreed well with other studies done with similar emissions sources. The corresponding correlation plots of toluene and benzene for those two regions can be found in Figure S3.

And we removed this description to the case studies because we included the OH exposure to help discuss the air mass age.

**Page 14, Figure 4: Units for j_NO2 and OH exposure are missing. Same for Figure 5.**

The units were added in the figures of the revised manuscript.

**Page 14, line 366: "As mentioned before, photochemical oxidation contributed a large fraction to acetone and the larger aliphatic carbonyls over the Arabian Gulf and Suez areas" I think this statement needs stronger support. The terrestrial biosphere may emit a large amount of acetone. Moreover, the acetone yield from terpene oxidation is quite high, and the acetone production from terpenes may not be a huge source globally, but could well be in certain regions. Similarly, some of these compounds may be from anthropogenic sources or biomass burning.**

As the empirical calculation was removed, this conclusion previously derived from the results were removed as well. However, we would like to address the comment given by the reviewer as follows. The high mixing ratios of acetone precursors (propane and several higher alkanes) combined with strong photochemical processing, provide a strong regional secondary source. Regional biogenic sources of acetone are likely to be small given that much of the Arabian Peninsula is desert and little isoprene (a common biogenic emission) and monoterpenes was detected (most of the time below detection limit and the average was around tens of ppt). In the Arabian Gulf no significant increase of acetonitrile was observed so that a biomass burning

contribution to acetone can be ruled out. In the region of Suez, acetonitrile levels were somewhat elevated. The lowest of acetone to acetonitrile ratio was only during the biomass burning plum spikes in Suez, ranging from 7-10, which was similar to the ratios of acetone to acetonitrile reported by Holzinger et al. (2005) in aged biomass burning plume over the Eastern Mediterranean. During the rest of the time, the acetone to acetonitrile ratio was above 10 and up to 30. It suggested that the biomass burning source is still likely minor compared to the secondary source.

Holzinger, R., Williams, J., Salisbury, G., Klüpfel, T., de Reus, M., Traub, M., Crutzen, P. J., and Lelieveld, J.: Oxygenated compounds in aged biomass burning plumes over the Eastern Mediterranean: evidence for strong secondary production of methanol and acetone, Atmos. Chem. Phys., 5, 39-46, 10.5194/acp-5-39-2005, 2005.

**Page 14, line 374-376: The authors concluded that the carbonyls are co-produced via photochemical oxidation because they show strong correlation with ozone. I am not totally convinced. It is plausible that these carbonyls may be co-emitted with ozone precursors.**

We have added support for this statement by referencing the work of Tadic et al., (2020), who studied the production of O3 in the region in detail. In order to retain the reviewer's point we also note that primary emissions may also occur, as follows "Tadic et al. (2020) reported the net ozone production rate over the Arabian Gulf (32 ppb d$^{-1}$) was the greatest over the Arabian Peninsula. They show that strong ozone forming photochemistry occurred in this region, which would lead to abundant secondary photo-chemically produced products (including carbonyls). However, it should be noted the good correlation between ozone and carbonyls could in part be due to carbonyls co-emitted with ozone precursors (hydrocarbons) as primary emissions."

Tadic, I., Crowley, J. N., Dienhart, D., Eger, P., Harder, H., Hottmann, B., Martinez, M., Parchatka, U., Paris, J.-D., Pozzer, A., Rohloff, R., Schuladen, J., Shenolikar, J., Tauer, S., Lelieveld, J., and Fischer, H.: Net ozone production and its relationship to nitrogen oxides and volatile organic compounds in the marine boundary layer around the Arabian Peninsula, Atmospheric Chemistry and Physics, 20, 6769-6787, 10.5194/acp-20-6769-2020, 2020.

**Page 14, line 385: I noticed that the spatial distribution oil fields and refineries is from the year of 2007, while this cruise campaign was conducted in 2017. Is it possible to obtain more recent information on this?**

Unfortunately, we do not have access to any newer dataset of the oil and refineries distributions. Previously published work regarding the AQABA (Bourtsoukidis et al., 2019; Pfannerstill et al., 2019) all reported the distribution from 2007 data base.

**Page 15, line 405: "The biomass burning emissions were probably transported by on the prevailing northerly wind above Northeast Egypt…" is this supported by backward trajectory analysis or meteorological conditions?**

The prevailing wind direction in Suez was shown in Figure S1. We added "…by on the prevailing northerly wind (Figure S1) above Northeast Egypt…" in the revised manuscript.

**Page 16, Figure 6: this is interesting. how are these not-assigned ions calibrated/quantified?**

Where no calibration gas is available but a mass is detected then the mixing ratios of those ions were calculated based on an established theoretic calculation method (Lindinger et al. 1998) using a fixed proton transfer reaction rate constant ($k_{PTR}$) of $2.0 \times 10^{-9}$ cm$^3$ s$^{-1}$. Fortunately the protonation rate constant for all compounds is very similar, so that this approach yields reasonable results when direct in-field calibration is not possible.

Lindinger, W., Hansel, A., and Jordan, A.: On-line monitoring of volatile organic compounds at pptv levels by means of proton-transfer-reaction mass spectrometry (PTR-MS) medical applications, food control and environmental research, International Journal of Mass Spectrometry and Ion Processes, 173, 191-241, 1998.

**Page 17, line 449-452: any GC-FID measurements of isoprene during this period?**

Yes. The GC-FID also measured isoprene during AQABA campaign. However, as mentioned already in the text (line 417-418): According to the GC-FID measurement, isoprene was below the detection limit for most of the time during the AQABA cruise with the highest values observed in Suez (10 - 350 ppt).

**Page 17, line 460-462: "The model considers direct emissions (such as anthropogenic, biogenic, biomass burning etc.), atmospheric transport and mixing, photochemical production of carbonyls (by OH, O3 and NO3), and physical and chemical removal processes." This belongs to the method section where the model is introduced.**

We now moved this part to the method section 2.5.

**Page 17, line 465-466: "acetone was overestimated by a factor within 1.5 over the Arabian Sea, Gulf of Aden and Gulf of Oman, and underestimated by a factor within 2.5 over the other regions." Note that previous studies reported downward oceanic fluxes of acetone (ocean being a net sink) in the continental outflow from polluted regions (e.g., Marandino et al., 2005; Schlundt et al., 2017; Yang et al., 2014). How does the model treat the air-sea exchange of acetone? This key information is missing.**

The sub-model AIRSEA (Pozzer et al. 2006) implemented in EMAC calculates the exchange of acetone between the ocean and the atmosphere. The transfer velocity is calculated online. In general, ocean is a net sink for acetone, but regionally could be an emitter of acetone. This strongly depends on its concentration, however, away from sources the surface seawater is close to equilibrium with boundary layer air (Williams et al. 2004). For acetone in the model, a constant water concentration of 15 nmol/L is used, following the suggestion of Fischer et al. (2012). We added this information in the method part (section 2.5).

Fischer, E. V., Jacob, D. J., Millet, D. B., Yantosca, R. M., and Mao, J.: The role of the ocean in the global atmospheric budget of acetone, Geophysical Research Letters, 39, n/a-n/a, 10.1029/2011gl050086, 2012.

Pozzer, A., Jöckel, P., Sander, R., Williams, J., Ganzeveld, L., and Lelieveld, J.: Technical Note: The MESSy-submodel AIRSEA calculating the air-sea exchange of chemical species, Atmos. Chem. Phys., 6, 5435-5444, 10.5194/acp-6-5435-2006, 2006.

Williams, J., Holzinger, R., Gros, V., Xu, X., Atlas, E., and Wallace, D. W. R.: Measurements of organic species in air and seawater from the tropical Atlantic, Geophysical Research Letters, 31, 10.1029/2004gl020012, 2004.

**Page 19, line 504-505: "This indicates that the source of acetaldehyde was probably not from direct biogenic production…" this sentence is confusing. I think "direct oceanic acetaldehyde emissions are probably insufficient to explain…" might be more accurate?**

Here we wanted to emphasize that acetaldehyde is more related to a non-biogenic emission mechanism based on its poor correlation with DMS. We now correct the sentence to make it clearer: "This indicates that the direct biogenic acetaldehyde emissions from the ocean are probably insufficient to explain the measured acetaldehyde."

**Page 19, line 513-514: "To our knowledge, there is no clear experimental evidence showing the ocean to be a sink for acetaldehyde." This is probably true, but the language is vague (I think "direct" is a better choice than "clear" here), and several important studies are missing here. Schlundt et al. (2017) reported net downward fluxes of acetaldehyde in the polluted marine boundary layer (ocean is a net sink), which is inferred from measurements in the atmosphere and the surface ocean (so not "direct" evidence). Yang et al. (2014) reported oceanic fluxes of acetaldehyde using eddy covariance method (this is direct flux measurements). Indeed the fluxes were mostly upward for acetaldehyde (ocean is a net source), there appears to be a few points indicative of net downward fluxes, although are probably close to the limit detection of that system.**

Thanks for pointing out that several important studies were omitted when considering this point. We have added now them in the revised manuscript as follows: "… reporting the upper limit of the net ocean emission of acetaldehyde to be 34 Tg a$^{-1}$. Yang et al. (2014) quantified the air-sea fluxes of several OVOCs over Atlantic Ocean by eddy covariance measurements, showing ocean is a net source for acetaldehyde. Although Schlundt et al. (2017) reported uptake of acetaldehyde by the ocean from measurement-inferred fluxes in western Pacific coastal regions, to our knowledge, there is no direct experimental evidence showing the ocean to be a sink for acetaldehyde. "

**Page 21, line 549-554: can the authors please provide new plots like Figure 7 but for other major acetaldehyde/acetone precursors, such as ethane, propane, butanes, and perhaps ethanol as well? This would certainly make the case stronger, and may provide key insight into the acetaldehyde budget in this region. Even "remote" regions in this work are not really that far from the source regions. Therefore it remains unclear to what degree the underestimation/overestimation of carbonyls is due to their precursors. Emission inventories often have a hard time capturing the anthropogenic emissions, especially from the oil and gas industry.**

Exactly these precursor plots are already provided in the paper of Bourtsoukidis et al. (2019). Furthermore, Bourtsoukidis et al. (2020) already published a measurement-model comparison for the key precursors

(ethane, propane and butane). After implementing the new deep water source, the model was able to mostly reproduce the measurements of hydrocarbons over most areas except for the large underestimation over Suez and the Arabian Gulf, which was mentioned in the text. Therefore, we think it likely that other precursors which were not included in the model may contribute to the underestimation of acetaldehyde. Ethanol was not measured during the campaign. We do now include model comparison of alkenes (ethene and propene) to further support our argument of nighttime ozonolysis as a potential source in the Arabian Gulf as suggested by the other referee. In order to more clearly demonstrate the anthropogenic contribution to the model bias of acetaldehyde, we revised manuscript as follows:

Over the Arabian Gulf and Suez, the intensive photochemical production of carbonyls is apparent. Bourtsoukidis et al. (2020) compared measured hydrocarbons (ethane, propane, and butane) with the results from model simulations (the same model used in this study with the newly discovered deep water source implemented). The model was able to reproduce the measurement over most regions except for some significant model underestimations in Suez and Arabian Gulf, in which local and small-scale emissions were difficult for the model to capture. Therefore, an underestimation of the precursor hydrocarbons, as well as those large alkanes, alkenes and cyclic hydrocarbons which were not measured ($> C8$) or included in the model ($> C5$), could be a reason for the model underestimation of acetaldehyde especially in polluted regions. In addition, as mentioned in the previous case studies, high ozone mixing ratios were observed over the Arabian Gulf especially during the nighttime. Ethene and propene were found to be significantly underestimated during the nighttime high ozone period by a factor over 10 (Figure S9), which indicates that the nighttime ozonolysis of alkenes could be another important source for acetaldehyde, formaldehyde and other carbonyls (Atkinson et al., 1995;Altshuller, 1993) in the Arabian Gulf.

**Page 21, line 570: what biomass burning emission inventory is used in this work?**

For the biomass burning emissions, we used the global fire assimilation system emissions data from Kaiser et al. (2012). A sentence has been added in the method section 2.4 to address: The global fire assimilation system was used for biomass burning emissions (Kaiser et.al., 2012)."

Kaiser, J. W., Heil, A., Andreae, M. O., Benedetti, A., Chubarova, N., Jones, L., Morcrette, J.-J., Razinger, M., Schultz, M. G., Suttie, M., and van der Werf, G. R. (2012). Biomass burning emissions estimated with a global fire assimilation system based on observed fire radiative power. BG, 9:527-554

[revised manuscript text omitted]

---

## Author Comment (AC2) · 17 Jul 2020

Dear Referee,

Thank you very much for you thoughtful and constructive comments for our work. We addressed them in the revised version of the manuscript as described below:

**Overview:**

**The manuscript presented ship-borne measurements of carbonyl compounds (carbonyls) around the Arabian Peninsula during the AQABA campaign (end of June to end of August 2017) and interpreted the measurement with a chemical transport model (EMAC). Analyses were focused on 3 classified carbonyl groups—aliphatic ($C_nH_{2n}O$), aromatic ($C_nH_{2n-8}O$), and unsaturated ($C_nH_{2n-2}O$)—in terms of their regional distribution, source characterization and partitioning, model-measurement comparison in 8 geographical regions, and finally an in-depth discussion on the missing sources of acetaldehyde (C2 aliphatic carbonyl).**

**The manuscript is well organized with clear methods, thorough analyses, and coherent arguments. The topic is of great importance given that carbonyls can provide key clues on reactive carbon source and chemistry as an oxygenated and stable group of VOCs, and acetaldehyde has drawn increased attention recently with growing evidence of its large missing sources. This manuscript therefore provided valuable information in advancing the knowledge of atmospheric reactive carbon in the Middle East. I support the publication of the manuscript once the following issues being addressed.**

Thank you for noting the key points and significance of the study.

1) **The title is interesting but over-weights the topic towards acetaldehyde and meanwhile lacks a reflection of the large portion of work done on the overview analysis of different carbonyls in this region as presented in the manuscript.**
On reflection, we agree. The title is now changed to "Measurements of carbonyl compounds around the Arabian Peninsula: overview and model comparison".

2) **L22. "A smaller but still significant discrepancy …", are you trying to say that the model-measurement discrepancy of other carbonyls is smaller comparing to that of acetaldehyde? This sentence is not clear and needs to be reorganized.**
We revised this sentence in the revised manuscript as "We compared the measurements of acetaldehyde, acetone and methyl ethyl ketone to global chemistry-transport model (EMAC) results. A significant discrepancy was found for acetaldehyde, with the model underestimating the measured acetaldehyde mixing ratio by up to an order of magnitude."

3) **L130. How did you deal with below LOD data?**
The following statement was added to make this clear: " … and $9 \pm 6$ ppt for methyl ethyl ketone (MEK) (Table S1). The data below LOD were excluded from the data set instead of giving zero."

4) **L134-140. There are other measurement techniques that can differentiate ketone and aldehyde at a shared mass, better to give an average or a range of in-situ reported ketone:aldehyde ratio (e.g., acetone:propanal ratio) to prove the dominance of ketone.**
We added this information now in the revised manuscript as "… ketones tend to have longer atmospheric lifetimes and higher photochemical yields than aldehydes as mentioned in the introduction. The ratio of measured propanal to acetone was 0.07 in the western Pacific costal region (Schlundt et al., 2017), 0.06 in an urban Los Angeles (Borbon et al., 2013) and 0.17 - 0.22 in oil &gas production regions (summarized by Koss et al., 2017)…."

5) **Figure2. The description of air mass characteristics is not clear. For example, what's the difference between "source nearby" vs "source nearby, aged", "remote, aged" vs "remote, clean"?**

We agree that the original description may confuse readers. Now in the revised manuscript we have simplified the labeling of the air mass characteristics by dividing them into source nearby (Suez, Arabian Gulf, Gulf of Oman and Red Sea North) and remote (Mediterranean Sea, Red Sea South, Gulf of Aden and Arabian Sea), which is consistent with what Bourtsoukidis et al. (2019) reported based on the NMHCs variability-lifetime results (b factor).

6) **L246-247. "Interestingly, the mean acetaldehyde mixing ratio (0.62 ± 0.59 ppb) over Suez was twice the level found over the Mediterranean Sea, whilst the acetone level was only slightly higher", why is this interesting? What are you trying to imply?**
The main message we would like to give is that the air in Suez region was influenced by local formation in addition to the transportation from Mediterranean Sea, since the mixing ratios do not simply scale, we agree that the original text was not clear. We have now rewritten the text as follows: "Another region where abundant aliphatic carbonyls were observed was Suez region. The air in this region was mainly influenced by nearby cities and marine transportation (ship emissions within the Suez Chanel) (Bourtsoukidis et al., 2019;Pfannerstill et al., 2019). Therefore abundant precursors were available in Suez region, producing more carbonyls regionally especially for shorter-lived compounds (formaldehyde and acetaldehyde). Besides the local-scale emissions and photochemical production contribution to the carbonyls over Suez, the longer lived carbonyls (e.g. acetone) could be also transported from the Mediterranean Sea (where acetone was high). Four-day back trajectories indicate the air reaching Suez region was mostly originated from Europe continent passing over the Mediterranean Sea (Bourtsoukidis et al., 2019)…."

7) **L247-248. "Besides the local-scale emission and photochemical production contribution to the acetone over Suez, the longer lived acetone could be also transported from the Mediterranean Sea (where acetone was high)". Acetone is quite similar in Mediterranean and Suez. "where acetone was high" seems to indicate that acetone source in Suez is dominated by transport rather than local production. We do not know the contribution of transport vs. local source to acetone in Suez region, so need to provide evidence or reorganize the sentence.**
Yes. It is correct that we do not know the contribution fraction of transport and local source to acetone. We reorganized the sentences as specified in the last point (6).

8) **L249-250. "Although the mean mixing ratios of aliphatic carbonyls over Suez were lower than those over the Arabian Gulf, larger variations were observed." I'm not seeing the higher variation over Suez than Arabian Gulf.**

We reorganized the sentence to avoid possible confusion and misunderstanding: "Although the mean mixing ratios of aliphatic carbonyls over Suez were much lower than those over the Arabian Gulf, the variations were still more significant than other regions (not including the Arabian Gulf, see Table 1)."

9) **L254-258. The arguments here are based on back trajectory. How about the proposed deep sea local source of ethane and propane in Bourtsoukidis et al. 2020? This was found not sufficient to explain the model bias of acetaldehyde later in the paper, but it worth being brought up here as one potential explanation of high carbonyls in the Red Sea North and also briefly mention what you found later in the model …**

We agree that by adding the deep water source information as well as the model comparison result would make the argument more complete. It is now written in the revised manuscript as "…while the sourthern part was more influenced by air from the northern part of the Red Sea mixed with the air masses from desertic areas of central Africa. Therefore, less primary precursors as well as carbonyls were transported to the sourhtern part of the Red Sea compared to the northern part. Moreover, the unexpected sources of hydrocarbons (ethane and propane) from Northern Red Sea deep water reported by Bourtsoukidis et al. (2020) would lead to higher carbonyl levels in the Northen part compared with the Sourthern part due to the additional precursors in the Red Sea North. However, acetaldehdye was still found to be significantly underestimated compared to the model results, even taking the deep-water source into consideration (section 3.3). This indicates that extra sources of acetaldehyde may exist, which will be disscussed in detail in section 3.4."

10) **Table 2. Are you summarizing only surface or tower measurement? Need to clarify.**

The content of Table 2 was also questioned by reviewer 1. We have now updated the table by including more relevant studies, done in open sea and coastal areas because AQABA measured the marine atmosphere for most of the time. We included the coordinates (latitude and longitude) and measurement sampling height (above sea level) of these studies.

**Too few examples of Oil & Gas region and Forest, add more.**

We now added more studies done related to Oil&Gas regions. However, although lots of studies reported hydrocarbons (NMHCs) from oil&gas region over the world, we could only find OVOCs results of oil&gas related studies done in the US. In terms of the forest category, we excluded together with other categories (urban and rural) because they were less related to AQABA ship campaign atmosphere.

The revised table is attached in the end of this response.

11) **L291, the title of Section 3.2 "Chemistry of aliphatic carbonyls", does not reflect the source characterization discussed in the two case studies in Section 3.2.2.**

The section 3.1 and 3.2 were reorganized as we removed one section which will be addressed in the next point. The section titles were changed to

**12) L303, Eq. (2). Varying formats of this equation have been widely applied but I have one question regarding its application in secondary OVOCs from aged air. Eq. (2) assumes that both precursor hydrocarbon and aliphatic carbonyls experienced the same degree of OH exposure (or $\Delta t$). This assumption is reasonable for fresh air mass, but might not be valid for aged air, in which carbonyls kept being produced along the way (depending on its precursor's lifetime) and thus experienced different degrees of OH exposure. So, for long-transported air masses, this could lead to an overestimated OH exposure, an underestimated carbonyl secondary production, and thus an underestimated fractional contribution from oxidation to the measured carbonyl mixing ratio.**

**Would this uncertainty affect your conclusion (P12, L334-336; P22, L615-618) that hydrocarbon oxidation plays a more important role in polluted regions than other less-polluted regions, given the potential low bias in the estimated contribution for less-polluted/more-remote regions?**

Thanks for pointing out the inappropriate application of the Eq. (2) we applied in our study to estimate the carbonyls produced by precursors especially in remote areas where the primary emissions were unknown and the air was aged. The empirical method was also questioned by reviewer 1, because photolysis, primary emissions, dilution effect from the background air and uptake by the land of ocean could influence the small carbonyls to a large extent, factors which were not considered at all in the empirical calculation. We agree that this section does not therefore have much scientific contribution to the whole manuscript. As we compared the measurement with a comprehensive global model in the manuscript, we removed the part regarding the empirical calculations and corrected relevant text in the revised manuscript.

However, we kept the content of estimating the OH exposure using hydrocarbon ratios for the polluted regions where primary emissions were strong and measured (Arabian Gulf and Suez). And we moved this description to the case studies because we included the OH exposure there to help discuss the air mass age.

**13) L308-312. Table S4. How is the yield being derived from MCM?**

We assumed the alkylperoxy radicals ($RO_2$) would mainly react with NO (high NO condition) and then sum up the yield of all the reaction pathways of hydrocarbons that could produce carbonyls. Now as we decided to remove the empirical calculation section, the Table S4 was also removed from the supporting information.

14) **L318. Did you use 0-5AM data as in Yuan et al. 2012? Clarify that.**
   **L319. How do your calculated emission ratios compare to literature reported values from similar sources?**

Thanks for pointing out this important detail. We now updated the results by using the benzene and toluene data during the night according to actinic flux data and compared the emission ratios to other studies from similar sources. As mentioned in point 12, we moved OH exposure calculation to the case studies. The text is now modified as follows:

For further insight, we focused on a time series of selected trace gases along with the correlations among them to better identify the sources of the major aliphatic carbonyls. Meanwhile, we calculated the OH exposure ($[OH]\Delta t$) based on hydrocarbon ratios (Roberts et al., 1984;de Gouw et al., 2005;Yuan et al., 2012) for these polluted regions to better understand the photochemical aging of the major carbonyls using the following equation:

$$[OH]\Delta t = \frac{1}{k_X - k_Y} \cdot (In \left.\frac{[X]}{[Y]}\right|_{t=0} - In \frac{[X]}{[Y]}) , \qquad \text{Eq. (1)}$$

where X and Y refer to two hydrocarbon compounds with different rates of reaction with the OH radical (k). For this study, we chose toluene ($k_{OH+toluene}$: 5.63E-12 cm$^3$ molecule$^{-1}$s$^{-1}$ ) and benzene ($k_{OH+benzene}$: 1.22E-12 cm$^3$ molecule$^{-1}$s$^{-1}$) (Atkinson and Arey, 2003), because both compounds were measured by PTR-ToF-MS at high frequency and these values showed a good agreement with values measured by GC-FID (Figure S2). The approach detailed by Yuan et al. (2012) was applied to determine the initial emission ratio $\left.\frac{[X]}{[Y]}\right|_{t=0}$ in those two regions by only including nighttime data of benzene and toluene. We obtained initial emission ratios (toluene to benzene ratio) of 1.38 for the Arabian Gulf and 2.12 for the Suez region. Koss et al. (2017) summarized the toluene to benzene ratios observed in various locations and showed that urban and vehicle sources tend to have higher toluene to benzene ratio (mean ~ 2.5) than the ratios of oil & gas sources (mean ~ 1.2). Therefore, the toluene to benzene ratios obtained for those two regions agreed well with other studies done with similar emissions sources. The corresponding correlation plots of toluene and benzene for those two regions can be found in Figure S3."

15) **L329, "In general, the direct oxidation fraction varied from area to area for C1 to C3 carbonyls (formaldehyde, acetaldehyde and acetone)." Give the range of fractions for each species.**

We removed the part regarding the empirical calculations and corrected relevant text in the revised manuscript.

16) **L414, Section 3.3 title "air chemistry of unsaturated carbonyls" does not reflect the discussion in this section, which is more focused on potential precursor/source of large unsaturated carbonyls.**

We corrected the title to "Potential precursors and sources of unsaturated carbonyls" in the revised manuscript.

17) **L470-471. "A strong natural non methane hydrocarbon source from deep water in the Northern Red Sea was implemented in the model (Bourtsoukidis et al., 2020)." This needs to be added in the method section where you describe the model configuration. There you have stated that the**

**configuration is following Bourtsoukidis et al. 2020, but also worth pointing out that this newly proposed source is applied.**

Thanks for the suggestion. We now add this information to the method section 2.5 as "The model configuration in the study is the same as the model applied in Bourtsoukidis et al. (2020), where an extra natural non-methane hydrocarbon source (ethane and propane) was implemented."

18) **L520. "the air-sea submodel (Pozzer et al., 2006) was augmented to include …" what is the air-sea submodel? Is that a model coupled into the EMAC model? Or it's a separate model you used to get the ocean acetaldehyde concentration field? Need to clarify. And if it's coupled to EMAC, need to add in the EMAC model configuration section.**

This was also mentioned by reviewer 1. The AIRSEA submodel was coupled into the EMAC. We now added this information in the method section 2.5: "The exchange of organic compounds between ocean and atmosphere was considered in EMAC via AIRSEA submodel, described in detail in Pozzer et al. (2006). The transfer velocity is calculated online and the concentration in the water is prescribed by the user. For acetone, a constant water concentration of 15 nmol/L is used, following the suggestion of Fischer et al. (2012)."

19) **P21, Section 3.5.3, "Anthropogenic primary sources". Comparing to Section 3.5.1 and 3.5.2, this section is lack of analysis-based discussion.**

**Also, if the revised version kept the same length or longer, better to separate into two paragraphs, with the first one discussing the underestimated anthropogenic primary emissions of acetaldehyde precursors (L550-558) and the second talking about underestimated anthropogenic primary emissions of acetaldehyde itself (L558-end).**

We reorganized this section as suggested in the revised manuscript. In terms of the analysis-based discussion, we included one more plot in the supporting information (Figure S9. Time series of ozone mixing ratios and measurement to model ratios of acetaldehyde, propene and ethene over the Arabian Gulf ) inspired by the referee's later comments to further address the potential contribution to carbonyls from nighttime ozonolysis of alkenes.

**L551. Does the carbon number of unmeasured hydrocarbons start from C12? Figure 3 and 6 only considered HCs up to C8.**

Thanks for spotting this mistake. We corrected C12 to C8.

**L552-555. "Bourtsoukidis et al. (2020) compared measured hydrocarbons (ethane, propane, ethene etc.) with the results from model simulations (the same model used in this study) and periodically found significant model underestimation in both regions. This indicates that not all sources were present in the model's emission inventory". This seems to argue that the low bias of ethane, propane, ethene, etc. in the model inventory is one reason for the model underestimation of acetaldehyde, which contradicts the argument (P13, L340-349) based on Figure 3. The argument of non-measured and non-modeled HCs before this sentence is fine, but this sentence makes it unclear.**

As the Figure 3 related to empirical calculations of carbonyls was removed from the revised manuscript, there should be no confusion. We modified the sentences to make it clearer to demonstrate the

contribution from underestimation of precursors in the revised manuscript: "Bourtsoukidis et al. (2020) compared measured hydrocarbons (ethane, propane, and butane) with the results from model simulations (the same model used in this study with deep water source implemented). The model was able to mostly reproduce the measurement over different regions expect for periodically significant model underestimations in Suez and Arabian Gulf, in which local and small-scale emissions were difficult for model to capture. Therefore, an underestimation of the precursor hydrocarbons as well as those large alkanes, alkenes and cyclic hydrocarbons which were not measured (> C8) or included in the model (> C5) could be a reason for the model underestimation of acetaldehyde especially in polluted regions."

**L555-550. 1) need to say that the high ozone in case studies was observed during "nighttime". 2) Does the modeled occasional underestimate of ethene persist in nighttime too?**

1) We corrected the sentence to "As mentioned in the previous case studies, high ozone mixing ratios were observed over the Arabian Gulf and Suez especially during the nighttime."

2) Thanks for mentioning the alkene model comparison. We now made a new plot included in the Supporting Information showing the time series of measured to model ratios of alkenes together with ozone. Relevant text was also added in the revised manuscript: "Ethene and propene were found to be significantly underestimated during the nighttime high ozone period by a factor over 10 (Figure S9), which indicates that the nighttime ozonolysis of alkenes could be another important source for acetaldehyde, formaldehyde and other carbonyls (Atkinson et al., 1995;Altshuller, 1993) in the Arabian Gulf."

**L565-567. "Therefore, the active petroleum industry located in the Arabian Gulf and intensive marine transportation in Suez are likely primary sources of acetaldehyde and other carbonyls which were not well constrained in the model." This is not convincing.**

We have rewritten the sentence to raise the point without indicating certainty as follows: A possible explanation for the measurement-model discrepancy is that the active petroleum industry located in the Arabian Gulf and intensive marine transportation in Suez are primary sources of acetaldehyde and other carbonyls which were not well constrained in the model.

20) **L587, "Another acetaldehyde formation pathway …" suggest starting a new paragraph to discuss pyruvic acid.**

We started a new paragraph to discuss pyruvic acid in the revised manuscript.

**L590-592. "Zhou and Mopper (1997) pointed out that the net exchange direction for pyruvic acid is expected to be from the air to the sea due to its high partition coefficient (high solubility). Therefore, only low levels of pyruvic acid would be expected in the remote marine boundary layer". High air-to-sea partition tendency does not guarantee low level of pyruvic acid in the air-side, if gas-liquid partitioning is the only limiting process, its concentration in the air also depends on its degree of saturation in the surface seawater. A review of surface seawater concentration of pyruvic acid and an air-sea equilibrium calculation might be helpful.**

**L597-598. "As the air-sea exchange of pyruvic acid is limited, low levels of pyruvic acid were expected" Not clear what this means. How did you know that the air-sea exchange of pyruvic acid is limited? And why would that result in low levels of pyruvic acid?**

As the two comments above are related, we respond to them together as follows. Regarding the potential contribution of pyruvic acid photolysis to acetaldehyde, we now adopt the reviewers suggestion to make this point clear in the revised manuscript as follows: "…Pyruvic acid has been also observed in seawater (Kieber and Mopper, 1987;Zhou and Mopper, 1997) and was found up to 50 nM in the surface water of eastern pacific Ocean (Steinberg and Bada, 1984), while acetaldehyde was not the major product of aqueous-phase photolysis of pyruvic acid (Griffith et al., 2013). Zhou and Mopper (1997) pointed out that the net exchange direction for pyruvic acid is expected to be from the air to the sea due to high solubility, with Henry's law constant of $3.1 \times 10^3$ mol m$^{-3}$ Pa$^{-1}$ (Sander, 2015). Moreover, partitioning to aerosols could be an important sink for pyruvic acid (Reed et al.,2014; Griffith et al., 2013) : an increasing concentration trend of pyruvic acid was observed in marine aerosols over western North Pacific Ocean (Boreddy et al., 2017). Therefore, due to limited terrestrial biogenic sources of pyruvic acid for AQABA campaign, gas-phase level of pyruvic acid was expected to be low. Limited studies reported pyruvic acid level in marine boundary layer, Baboukas et al. (2000) measured $1.1 \pm 1.0$ ppt of pyruvic acid above the Atlantic Ocean….."

New literature citation:

Steinberg, S. M., and Bada, J. L.: Oxalic, glyoxalic and pyruvic acids in eastern Pacific Ocean waters, Journal of Marine Research, 42, 697-708, 10.1357/002224084788506068, 1984.

Sander, R.: Compilation of Henry's law constants (version 4.0) for water as solvent, Atmospheric Chemistry and Physics, 15, 4399-4981, 10.5194/acp-15-4399-2015, 2015.

Reed Harris, A. E., Ervens, B., Shoemaker, R. K., Kroll, J. A., Rapf, R. J., Griffith, E. C., Monod, A., and Vaida, V.: Photochemical kinetics of pyruvic acid in aqueous solution, J Phys Chem A, 118, 8505-8516, 10.1021/jp502186q, 2014.

**L598-599. "Even if we fully assign the m/z 89.024 to pyruvic acid, the contribution to acetaldehyde via photolysis of pyruvic acid is negligible compared other sources" Clarify why it is negligible, using 58 ppt and a 100% yield? Is this contribution detectable by instrument?**

We added detailed information to clarify this point in the revised manuscript as follows: "For the AQABA PTR-ToF-MS data set, enhanced signals were observed at m/z 89.024 with the mean mixing ratio of 35-110 ppt over different regions (Table S4), which is much more abundant than reported pyruvic acid levels by Baboukas et al. (2000). This might be due to the uncertainty associated with the theoretical methods of quantification used here or the presence of isomeric compounds on that mass, since pyruvic acid was not calibrated with the standard. Even if we assume the m/z 89.024 to be entirely pyruvic acid, with 60% yield of acetaldehyde via photolysis (IUPAC, 2019), it gave maximum 13 ppt of acetaldehyde over Arabian Gulf, 5-9 ppt over other regions, which were only 0.8% - 6% of the mean mixing ratios (Table S4). Detailed information of the calculation can be found in the Supporting Information. Therefore, we conclude that the contribution from the photolysis of pyruvic acid is not an important source for the unattributed acetaldehyde during the AQABA campaign."

New citation:

IUPAC Task Group on Atmospheric Chemical Kinetic Data Evaluation, (http://iupac.pole-ether.fr).

Accordingly, we added the detailed information of the acetaldehyde calculation in the supporting information as follow:

**Calculation of acetaldehyde yield from pyruvic acid photolysis**

In order to verify the contribution from the photolysis of pyruvic acid to acetaldehyde, we calculated the expected acetaldehyde produced through pyruvic acid photolysis over different regions assuming: (1) *m/z* 89.0234 is fully assigned to pyruvic acid; (2) the loss of pyruvic acid is only through photolysis; (3) 60% is the yield of acetaldehyde via pyruvic acid photolysis recommended by IUPAC (2019); (4) the loss of acetaldehyde is only through OH oxidation. The acetaldehyde produced via pyruvic acid photolysis can be calculated using following equation (consecutive reactions):

$$[Acetaldehyde] = [Pyruvic\ acid]\frac{J_{PA}}{k_{OH}[OH]-J_{PA}}[\exp(-J_{PA}\Delta t) - \exp(-k_{OH} \times [OH]\Delta t)] \qquad \text{Eq. S1}$$

$[Pyruvic\ acid]$ is the mean of pyruvic acid mixing ratio in each region. $J_{PA}$ represents the mean photolysis rate constant of pyruvic acid during the daytime (dawn to dusk) in each region calculated from the wavelength resolved actinic flux data using quantum yield of 0.2 as suggested by IUPAC (2019). The $k_{OH}$ is the rate constant of acetaldehyde reacting with OH radical ($1.5 \times 10^{-11}$ cm$^3$ molecule$^{-1}$s$^{-1}$, Table S3). The $[OH]$ concentrations in each area were the mean values during the daytime obtained from the EMAC model. The maximum acetaldehyde level as well as the corresponding time ($\Delta t$) can be derived from Eq. S1 as $\Delta t$ is the only variable. The results are shown in Table S4.

Table S4. Mean photolysis rate constant of pyruvic acid, OH concentrations, relative time ($\Delta t$) needed to reach the maximum acetaldehyde yield from pyruvic acid photolysis, maximum acetaldehyde and its fraction accounting the mean level over regions.

| Regions | $J_{PA}$ (s$^{-1}$) | OH (molecules cm$^{-3}$) | m/z 89.0234 pyruvic acid (H$^+$) (ppt) | $\Delta t$ (h) | Acetaldehyde maximum (ppt) | Fractions (%) |
|---|---|---|---|---|---|---|
| MS | $3.51 \times 10^{-5}$ | $6.52 \times 10^6$ | $39 \pm 8$ | 5.6 | 5.6 | 1.85 |
| SC | $3.44 \times 10^{-5}$ | $7.42 \times 10^6$ | $42 \pm 9$ | 5.2 | 5.3 | 0.85 |
| RSN | $3.52 \times 10^{-5}$ | $7.14 \times 10^6$ | $35 \pm 14$ | 5.2 | 4.7 | 0.92 |
| RSS | $3.00 \times 10^{-5}$ | $8.74 \times 10^6$ | $61 \pm 15$ | 4.9 | 6.2 | 1.98 |
| GA | $3.11 \times 10^{-5}$ | $7.20 \times 10^6$ | $57 \pm 12$ | 5.5 | 6.8 | 3.60 |
| AS | $2.74 \times 10^{-5}$ | $4.35 \times 10^6$ | $59 \pm 12$ | 7.8 | 9.4 | 5.88 |
| GO | $3.31 \times 10^{-5}$ | $7.89 \times 10^6$ | $65 \pm 10$ | 5.0 | 7.6 | 2.91 |
| AG | $3.29 \times 10^{-5}$ | $7.81 \times 10^6$ | $110 \pm 53$ | 5.1 | 12.9 | 0.75 |

**Minor/technical comments:**

**L17: typo "3,8"**

It is now corrected to "3.8" in the revised manuscript.

**L89: section numbering error, should start with 2.1**

It is now corrected to "2.1" in the revised manuscript.

**L131 and Table S1: 3σ in text but 1σ in Table S1**

We corrected Table S1 to 3σ.

**L135: add "of" between "number" and "three"**

It is now added in the revised manuscript as "… with a carbon number of three…)

**L156: "interaction with ocean/land" includes "dry deposition"**

We deleted "dry deposition" in the revised manuscript.

**Table 1. L193, L207. "Aliphatic CCs, Aromatic CCs, Unsaturated CCS", "CC" hasn't been defined.**

We changed "CCs" in the table to "Carbonyls" in the revised manuscript.

**Table 1. L193, L207. Extend the name of "S" to "Suez"**

We made this change in the revised manuscript.

**L348. "section" to "sections"**

The related content was deleted. Therefore, this correction was not applicable.

**Figures 4, 5, and 7. Change x-axis label from "Dateandtime" to "Dateandtime (UTC)", or clarity in the figure caption.**

We made this change in the revised manuscript.

**Figures 4 and 5. missing units for several y-axis, "J_NO2", "[OH]t", "Wind_D"**

We made this change in the revised manuscript.

**Figure 7a left panel. "S" to "Suez"**

We made this change in the revised manuscript.

**Figure S6. Only 6 regions shown in the figure.**

We added the other two regions in the Figure S6 in the revised manuscript.

[revised manuscript text omitted]

---

## Author Response (AR2)

Dear Dr. Robert Harley,

Thank you very much for your decision on our paper. We made the correction about how to deal with the data below
the limits of detection and addressed them as below. Meanwhile, data availability was updated in the revised manuscript
as "The data used in this study are available to all scientists agreeing to the AQABA protocol at:
http://doi.org/10.5281/zenodo.3974228 (Wang et al., 2020)."

**"Please can you consider one further point relating to handling of data below the limit of detection (LOD):**
**rather than excluding those points (which are still valid measurements) as you have done, is there a way you can**
**use some small non-zero value such as LOD/2 or the square root of the LOD or ..., to avoid biasing the mean by**
**systematically excluding all of the lowest measured values."**

Thanks for the comment and suggestion. We now included the data below the limit of detection using actual values
determined by the data processing method as suggested by the reviewer. The sentence was revised in the text (2.2.3)
as "Data below LOD were kept as determined for statistic analysis (Figure 2 and Table 1)." Table 1 and Figure 2 were
updated by including the data below the LOD. Following text were also revised accordingly (revised parts were marked
in red):

**Line 18**: acetaldehyde (0.16 ppb over the Arabian Sea… " Revised to "acetaldehyde (0.13 ppb over the Arabian Sea… ".

**Line 219-222**: "Acetaldehyde was measured at relatively low mixing ratios over the Arabian Sea (0.16 ± 0.12 ppb,
median: 0.13 ppb), which is comparable than the levels reported by the measurements done in northern-hemisphere
open ocean (see, Table 2). Over the Gulf of Aden, acetone and MEK had slightly higher mixing ratios than those over
the Arabian Sea." Revised to "Acetaldehyde was measured at relatively low mixing ratios over the Arabian Sea (0.13
± 0.12 ppb, median: 0.09 ppb), which is comparable than the levels reported by the measurements done in northern-
hemisphere open ocean (see, Table 2). Over the Gulf of Aden, acetaldehyde, acetone and MEK had slightly higher
mixing ratios than those over the Arabian Sea.".

**Line 381-383:** "The mixing ratios of unsaturated carbonyls were generally low with values below 30 ppt over the
Mediterranean Sea and the clean 381 regions (the Arabian Sea and the Gulf of Aden, 12 - 21 ppt). The Red Sea region
and the Gulf of Oman had slightly higher levels 382 (13 – 60 ppt). The highest values were again observed in the
Arabian Gulf (25 – 115 ppt) followed by Suez (11 – 68 ppt)." Revised to "The mixing ratios of unsaturated carbonyls
were generally ~ 10 ppt or lower than the LOD over the Mediterranean Sea and the clean regions (the Arabian Sea and
the Gulf of Aden). The Red Sea region and the Gulf of Oman had slightly higher levels (LOD – 40 ppt). The highest
values were again observed in the Arabian Gulf (20 – 110 ppt) followed by Suez (LOD – 60 ppt).".

[revised manuscript text omitted]

Table 1 Continued

| | | Aromatic Carbonyls | | | Unsaturated Carbonyls | | | | | |
|---|---|---|---|---|---|---|---|---|---|---|
| | | C7H6O | C8H8O | C9H10O | C4H6O | C5H8O | C6H10O | C7H12O | C8H14O | C9H16O |
| MS | mean | 0.02 | 0.01 | < LOD | 0.01 | 0.01 | 0.01 | < LOD | < LOD | < LOD |
| | SD | 0.03 | 0.01 | n.a. | 0.02 | 0.01 | 0.01 | n.a. | n.a. | n.a. |
| | median | 0.02 | 0.01 | < LOD | 0.01 | 0.01 | 0.01 | < LOD | < LOD | < LOD |

| Site | | 1 | 2 | 3 | 4 | 5 | 6 | 7 | 8 | 9 |
|---|---|---|---|---|---|---|---|---|---|---|
| Suez | mean | 0.09 | 0.03 | < LOD | 0.06 | 0.04 | 0.03 | 0.01 | 0.01 | < LOD |
| | SD | 0.20 | 0.04 | n.a. | 0.08 | 0.04 | 0.03 | 0.01 | 0.01 | n.a. |
| | median | 0.02 | 0.01 | < LOD | 0.04 | 0.02 | 0.02 | 0.01 | < LOD | < LOD |
| RSN | mean | 0.09 | 0.05 | 0.02 | 0.03 | 0.03 | 0.04 | 0.02 | 0.01 | 0.01 |
| | SD | 0.10 | 0.06 | 0.02 | 0.02 | 0.03 | 0.04 | 0.02 | 0.01 | 0.01 |
| | median | 0.06 | 0.04 | 0.01 | 0.02 | 0.03 | 0.03 | 0.01 | 0.01 | < LOD |
| RSS | mean | 0.05 | 0.04 | 0.03 | 0.01 | 0.02 | 0.02 | 0.01 | 0.03 | 0.01 |
| | SD | 0.06 | 0.06 | 0.03 | 0.01 | 0.02 | 0.02 | 0.01 | 0.07 | 0.01 |
| | median | 0.01 | 0.01 | 0.03 | 0.01 | 0.01 | 0.01 | < LOD | < LOD | < LOD |
| GA | mean | 0.02 | 0.02 | 0.01 | 0.01 | 0.01 | 0.02 | 0.01 | < LOD | < LOD |
| | SD | 0.03 | 0.02 | 0.01 | 0.01 | 0.01 | 0.01 | 0.01 | n.a. | n.a. |
| | median | 0.01 | 0.01 | 0.01 | 0.01 | 0.01 | 0.01 | < LOD | < LOD | < LOD |
| AS | mean | 0.02 | 0.01 | < LOD | 0.01 | 0.01 | 0.01 | < LOD | < LOD | < LOD |
| | SD | 0.03 | 0.01 | n.a. | 0.01 | 0.01 | 0.01 | n.a. | n.a. | n.a. |
| | median | 0.01 | 0.01 | < LOD | 0.01 | 0.01 | 0.01 | < LOD | < LOD | < LOD |
| GO | mean | 0.04 | 0.04 | 0.02 | 0.02 | 0.02 | 0.02 | 0.01 | 0.01 | 0.01 |
| | SD | 0.06 | 0.05 | 0.03 | 0.01 | 0.01 | 0.01 | 0.01 | 0.01 | n.a. |
| | median | 0.02 | 0.02 | 0.01 | 0.02 | 0.02 | 0.02 | 0.01 | 0.01 | < LOD |
| AG | mean | 0.12 | 0.13 | 0.04 | 0.07 | 0.11 | 0.12 | 0.05 | 0.03 | 0.02 |
| | SD | 0.14 | 0.10 | 0.04 | 0.06 | 0.10 | 0.10 | 0.05 | 0.03 | 0.02 |
| | median | 0.08 | 0.10 | 0.03 | 0.04 | 0.07 | 0.09 | 0.04 | 0.03 | 0.01 |

< LOD: the mixing ratios were lower than the limit of detection.

n.a.: not available

[revised manuscript text omitted]